# LithoDreamer: A Physics-Informed World Model for Multi-Stage Computational Lithography

Yuqi Jiang [1]  Yumeng Liu [1]  Zimu Li [1]  Jinyuan Deng [1]  Qian Jin [1]  Yucheng Cui [1]  Yu Li [1]  Xunzhao Yin [1]  Qi Sun [1]  Cheng Zhuo [1]

## Abstract

As semiconductor technology nodes scale, computational lithography is essential for ensuring yield and performance. However, lithography is a continuous physical process involving mask optimization, optical imaging, resist exposure, and development, which existing models fail to capture. To overcome this limitation, we present *LithoDreamer*, the first physics-informed World Model (WM) framework for computational lithography, which formulates the "Layout-Mask-Resist Image-After Development Image (ADI)" pipeline as a decision-driven multi-step evolution system. LithoDreamer captures feature changes between adjacent states to model stage-specific physics-informed latent spaces, in which it controls process intervention exploration and drives subsequent state transitions. To achieve interpretable intervention optimization without continuous supervision, we propose a contrastive variational optimization paradigm that contrasts the latent differences between intervention paths with variational evolution constraints, guiding the model to generate evolutions consistent with real lithography physics. Experiments show LithoDreamer achieves state-of-the-art performance in forward evolution and inverse planning. Our lithography dataset is publicly available at GitHub ⭘.

## 1. Introduction

With the continuous advancement of advanced technology nodes, lithography has become the core bottleneck restricting chip manufacturing yield. Computational lithography, through analytical modeling of mask design, optical imag-

[1]Zhejiang University, No. 38 Zheda Road, Xihu District, Hangzhou, Zhejiang Province, China. Correspondence to: Qi Sun <qisunchn@zju.edu.cn>.

*Proceedings of the $43^{rd}$ International Conference on Machine Learning*, Seoul, South Korea. PMLR 306, 2026. Copyright 2026 by the author(s).

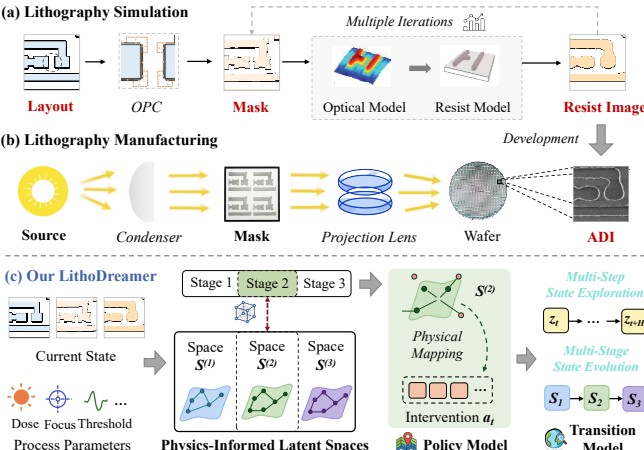

*Figure 1.* Comparison of the different processes: (a) Typical commercial simulation workflow; (b) Actual physical lithography manufacturing process; (c) The evolution workflow of our LithoDreamer's process intervention and lithography state.

ing, and photoresist reactions, simulates the practical manufacturing process and plays a key role in addressing imaging complexities and manufacturing constraints (Yang & Ren, 2023; Jin et al., 2025b; Jiang et al., 2026). However, in Figure 1, the lithography process involves multiple physical principles and is influenced by layout structures, mask geometries, and manufacturing process variations. These make lithography modeling and optimization face huge challenges in multi-stage (i.e., "Layout-Mask-Resist Image-After Development Image (ADI)") physical evolution.

Traditional Optical Proximity Correction (OPC) and Inverse Lithography Technology (ILT) methods (Chen et al., 2025; Yang et al., 2025; Jiang et al., 2024) treat the lithography process as a goal-driven numerical optimization problem, gradually modifying the mask in an explicit variable space by repeatedly invoking physical simulators to meet imaging and process constraints. These methods rely on complex physical and optical computations, and the cost increases rapidly with the problem scale, which is difficult to optimize in large designs efficiently. In recent years, machine learning methods (Jin et al., 2025a; Wang et al., 2025; Jiang et al., 2025; Jin et al., 2025c) have been proposed, which fit large

amounts of data to learn point-to-point mappings from layout and process conditions to mask, resist image, and ADI representations. However, such methods define the lithography process as static predictions and cannot explicitly model continuous process interventions that drive lithography state evolution in real-world environments. Therefore, they struggle to address the need for continuous process adjustments in the multi-stage optimization scenarios.

Recently, World Models (WMs) (Zhao et al., 2025; Bar et al., 2025; Alonso et al., 2024; Huang et al., 2025) have been dedicated to learning long-term decision-making and planning in dynamic scenarios. By simulating the interaction between agents and the environment, WMs can capture the trajectories of state changes under continuous action interventions. This ability has proven effective in autonomous driving and embodied intelligence. Similarly, the lithography process requires a series of process interventions to adjust the lithographic environment and characterize how these interventions gradually affect the states of the mask, resist image, and ADI. These representations provided by WMs offer an efficient modeling framework for lithographic process planning and multi-stage state evolution.

However, directly applying existing WM frameworks to computational lithography still faces fundamental challenges. First, lithography involves multiple stages with distinct physical properties, making it difficult to capture stage-dependent dynamics within a single latent model. Second, while process conditions such as dose, focus, source, and threshold are observable, the fine-grained interventions that drive continuous pattern evolution are typically unobserved and lack direct supervision, hindering the learning of intervention-conditioned dynamics.

To overcome these challenges, we propose the first WM model for computational lithography called **LithoDreamer**, designed to unify the multi-stage **"Layout-Mask-Resist Image-ADI"** into a physics-inspired continuous evolution system. Our contributions are as follows:

- We propose the first physics-informed lithography WM, LithoDreamer, which treats the lithography pipeline as a multi-stage physical evolution system, enabling lithography to be modeled as a causal and decision-driven process rather than a static forward prediction.

- We design a Space Prior Approximation (SPA) method that characterizes stage-specific physical latent spaces from statistical state variations, constraining interventions along physically consistent evolution directions.

- We introduce a contrastive variational optimization paradigm that applies variational inference and contrastive learning to jointly explore the optimal solutions for process intervention planning and state transitions without discrete actions.

- We construct a dataset of 280K paired samples. LithoDreamer achieves EPEs of 0.96/1.06 forward evolution and 0.89/0.97 inverse planning for in-domain (ID) and out-of-domain (OOD), demonstrating superior performance and generalization.

## 2. Preliminaries

### 2.1. Lithography and Dataset

We study optical lithography, where printed patterns are governed by process-dependent variations. The lithography process is parameterized by exposure dose (exposure energy), focus (defocus offset), threshold (photoresist development threshold), and source (illumination source distribution), which jointly define a multi-dimensional process condition. Under each process condition, pattern formation follows a structured **multi-stage** pipeline, i.e., "Layout-Mask-Resist Image-ADI". Within each stage, the lithography state undergoes a continuous latent evolution driven by process-dependent physical effects, which we model as **multi-step** state evolutions toward stage-consistent outcomes.

Each data sample corresponds to a specific layout under a specific process-parameter configuration (i.e., a combination of dose, focus, threshold, and source). The dataset comprises 280k paired samples derived from industrial-grade commercial lithography data collected from a 55 nm manufacturing line, encompassing diverse layouts under varied configurations. For each sample, a local region is cropped from the global layout with a fixed physical field-of-view of 6000 nm × 6000 nm and rasterized into 512 × 512 pixels.

### 2.2. Related Work on Computational Lithography

Early computational lithography methods are mainly physics-based, with OPC and ILT as representative approaches. For example, Chen et al. (Chen et al., 2025) combine optical imaging models with empirical rules to locally adjust mask geometries for proximity-effect compensation. L2O-ILT (Zhu et al., 2023) formulates lithography as an inverse problem and optimizes mask structures in a continuous shape space. However, these methods rely on accurate physical modeling and computationally expensive simulations in industrial applications. Recent learning-based methods reduce such reliance by learning mappings between lithography representations. Unitho (Jin et al., 2025a) uses Transformers to model multi-stage mappings among Layout, Mask, and Resist Image, while LMLitho (Wang et al., 2025) introduces large models to represent diverse process conditions better. Nevertheless, existing methods mostly focus on isolated stages or single-step forward mappings, without explicitly modeling cross-stage state evolution under continuous process interventions. This limits their ability to capture lithography dynamics in a unified manner.

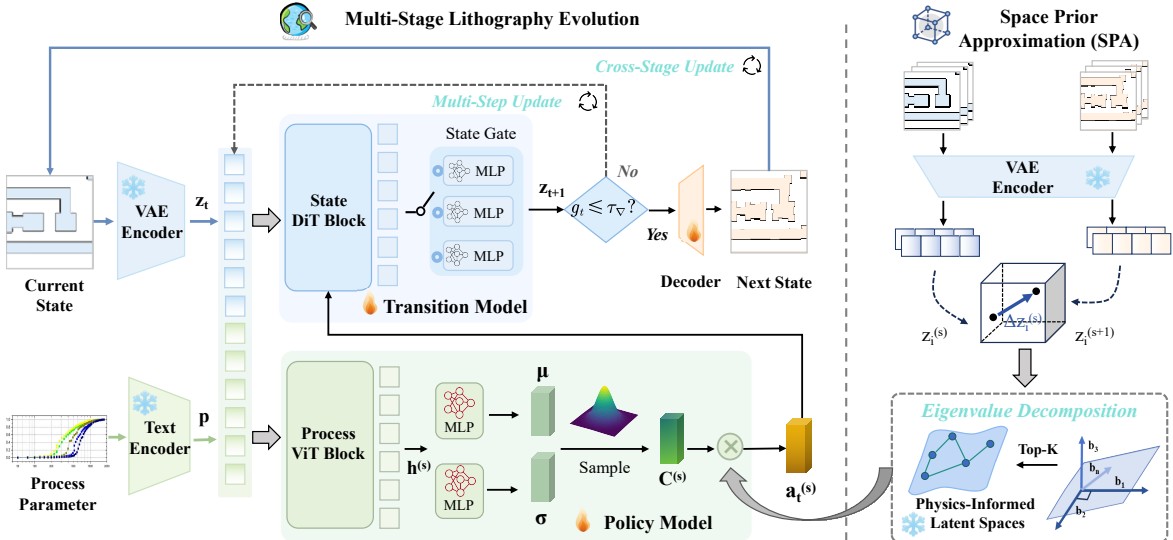

*Figure 2.* Overview of the LithoDreamer framework.

## 2.3. Applications of World Models

WMs learn latent environment dynamics from historical observations, enabling multi-step state prediction and planning. This paradigm has been widely used in embodied intelligence and autonomous driving to support long-horizon reasoning and efficient decision-making. For example, DriveDreamer-2 (Zhao et al., 2025) combines LLMs with generative WMs for multi-view consistent autonomous driving modeling. NWM (Bar et al., 2025) formulates world modeling as conditional video generation to synthesize future observations for path planning and trajectory evaluation. However, existing WMs struggle to adapt to complex stage evolutions caused by continuous process interventions in computational lithography. So we are committed to designing a WM modeling framework for computational lithography, aimed at supporting multi-stage and multi-step process modeling and state prediction within a unified framework.

## 3. Method

As shown in Figure 2, LithoDreamer consists of three components: stage-specific latent spaces for modeling lithography dynamics, the policy model for planning latent process interventions, and the transition model for predicting intervention-guided state evolution within and across stages.

### 3.1. Latent Spaces: Embedding of Physical Priors

The Layout-Mask, Mask-Resist Image, and Resist Image-ADI stages involve distinct process interventions that drive state transitions under stage-specific physical dynamics. To capture such evolution constraints, we propose Space Prior Approximation (SPA), which estimates a stage-specific basis matrix $\mathbf{B}^{(s)}$ to span the physics-informed latent space $\mathcal{S}^{(s)}$,

thereby constraining process interventions along physically consistent and feasible evolution directions.

Formally, let a frozen encoder $E(\cdot)$ map the state $\mathbf{x}^{(s)}$ at stage $s$ to a latent representation $\mathbf{z}^{(s)} = E(\mathbf{x}^{(s)}) \in \mathbb{R}^d$. For each adjacent-stage pair, we define the latent variation as:

$$\Delta \mathbf{z}_i^{(s)} = \mathbf{z}_i^{(s+1)} - \mathbf{z}_i^{(s)}, \tag{1}$$

where $i$ denotes the sample index. This vector represents the effective latent evolution direction governed by stage-specific physical dynamics. SPA estimates an orthonormal basis matrix $\mathbf{B}^{(s)}$ constructed from a set of $k$ basis vectors:

$$\mathbf{B}^{(s)} = [\mathbf{b}_1^{(s)}, \dots, \mathbf{b}_k^{(s)}] \in \mathbb{R}^{d \times k}, \quad (\mathbf{B}^{(s)})^\top \mathbf{B}^{(s)} = \mathbf{I}_k, \tag{2}$$

where $k \ll d$, and $\mathbf{I}_k$ denotes the $k \times k$ identity matrix. Using these basis vectors, the stage-specific physics-informed latent space $\mathcal{S}^{(s)}$ is then defined as:

$$\mathcal{S}^{(s)} = \left\{ \sum_{j=1}^{k} \alpha_j \, \mathbf{b}_j^{(s)} \,\middle|\, \alpha_j \in \mathbb{R} \right\}, \tag{3}$$

where $\alpha_j$ is the scalar coefficient associated with $\mathbf{b}_j^{(s)}$.

To identify $\mathbf{B}^{(s)}$, SPA requires each latent variation $\Delta \mathbf{z}_i^{(s)}$ to be well approximated by its projection onto $\mathcal{S}^{(s)}$. The orthogonal projection is:

$$\hat{\Delta} \mathbf{z}_i^{(s)} = \mathbf{B}^{(s)} (\mathbf{B}^{(s)})^\top \Delta \mathbf{z}_i^{(s)}. \tag{4}$$

Accordingly, $\mathbf{B}^{(s)}$ is estimated by minimizing the projection reconstruction error:

$$\min_{\mathbf{B}^{(s)}} \sum_{i=1}^{N_s} \left\| \Delta \mathbf{z}_i^{(s)} - \hat{\Delta} \mathbf{z}_i^{(s)} \right\|_2^2, \quad (\mathbf{B}^{(s)})^\top \mathbf{B}^{(s)} = \mathbf{I}_k, \tag{5}$$

where $N_s$ denotes the number of samples at stage $s$, set to 10k. Under the orthonormality constraint, the reconstruction error can be decomposed as:

$$\left\| \Delta \mathbf{z}_i^{(s)} - \hat{\Delta \mathbf{z}}_i^{(s)} \right\|_2^2 = \left\| \Delta \mathbf{z}_i^{(s)} \right\|_2^2 - \left\| \left( \mathbf{B}^{(s)} \right)^\top \Delta \mathbf{z}_i^{(s)} \right\|_2^2. \tag{6}$$

Since the first term is independent of $\mathbf{B}^{(s)}$, minimizing the reconstruction error is equivalent to maximizing the retained projection energy. Let $\tilde{\Delta \mathbf{z}}_i^{(s)}$ denote the mean-centered latent variation. We define the corresponding second-moment matrix as:

$$\mathbf{C}^{(s)} = \frac{1}{N_s} \sum_{i=1}^{N_s} \tilde{\Delta \mathbf{z}}_i^{(s)} \left( \tilde{\Delta \mathbf{z}}_i^{(s)} \right)^\top. \tag{7}$$

The above objective can then be written as $\mathrm{Tr}\left( \left( \mathbf{B}^{(s)} \right)^\top \mathbf{C}^{(s)} \mathbf{B}^{(s)} \right)$, whose maximizer is given by the top-$k$ eigenvectors of $\mathbf{C}^{(s)}$. These eigenvectors form $\mathbf{B}^{(s)}$ and span $\mathcal{S}^{(s)}$. Once estimated, $\mathbf{B}^{(s)}$ is fixed to constrain process interventions within physically consistent evolution directions.

## 3.2. Policy Model: Planning of Process Interventions

Given the current latent state and process-window parameters at stage $s$, the policy model plans latent process interventions within the physics-informed latent space $\mathcal{S}^{(s)}$. Rather than predicting a deterministic intervention, we adopt a stochastic policy that samples interventions from $\mathcal{S}^{(s)}$, thereby enabling exploration and uncertainty representation.

### 3.2.1. DISTRIBUTIONAL MODELING

Process interventions in computational lithography are underdetermined, as multiple intervention strategies may lead to similar outcomes under identical conditions. Accordingly, we model process interventions as random variables and learn their conditional distributions. At evolution step $t$ of stage $s$, the policy model $\pi_\theta$ takes the current latent state $\mathbf{z}_t^{(s)}$ and process-window parameters $\mathbf{p}$ as input. A Vision Transformer (ViT) extracts condition-dependent features, which are subsequently fed into a Multilayer Perceptron (MLP) to predict the parameters of the latent intervention coefficient distribution:

$$\mathbf{h}_t^{(s)} = \mathrm{ViT}_\theta \left( \mathbf{z}_t^{(s)}, \mathbf{p} \right), \quad \left( \boldsymbol{\mu}_t^{(s)}, \boldsymbol{\sigma}_t^{(s)} \right) = \mathrm{MLP}_\theta \left( \mathbf{h}_t^{(s)} \right), \tag{8}$$

where $\boldsymbol{\mu}_t^{(s)}, \boldsymbol{\sigma}_t^{(s)} \in \mathbb{R}^k$ denote the mean and standard deviation, and $k$ is the dimensionality of the stage-specific latent space. The intervention coefficients are sampled using the reparameterization trick, which enables the introduction of randomness in a differentiable manner. We first express the intervention coefficients as a deterministic function of the

learned mean and standard deviation, along with random noise drawn from a standard Gaussian distribution:

$$\mathbf{c}_t^{(s)} = \boldsymbol{\mu}_t^{(s)} + \boldsymbol{\sigma}_t^{(s)} \odot \boldsymbol{\epsilon}, \quad \boldsymbol{\epsilon} \sim \mathcal{N}(\mathbf{0}, \mathbf{I}), \tag{9}$$

where $\mathbf{c}_t^{(s)} \in \mathbb{R}^k$ denotes a sampled intervention coefficient vector at evolution step $t$ within stage $s$. By using the reparameterization trick, we ensure that the intervention coefficients $\mathbf{c}_t^{(s)}$ are differentiable with respect to the model parameters $\boldsymbol{\mu}_t^{(s)}$ and $\boldsymbol{\sigma}_t^{(s)}$. This allows us to optimize the intervention distribution by computing gradients through backpropagation. Through this distributional formulation, the policy model induces a stochastic intervention process that generates diverse intervention candidates in the latent space. As the intervention distribution is conditioned on the evolving latent state, it can be adaptively updated during state transitions, which facilitates multi-step planning and exploration within the stage-specific feasible space.

### 3.2.2. DIFFERENTIABLE MAPPING

The sampled intervention coefficient $\mathbf{c}_t^{(s)}$ does not necessarily correspond to a physically admissible intervention direction. To ensure the feasibility of process interventions, we map the coefficients into the physics-informed latent space $\mathcal{S}^{(s)}$. The final latent process intervention is obtained through a differentiable linear mapping:

$$\mathbf{a}_t^{(s)} = \mathbf{B}^{(s)} \mathbf{c}_t^{(s)}, \tag{10}$$

where $\mathbf{a}_t^{(s)}$ denotes the intervention in the latent space that aligns with the stage-specific dynamics, ensuring that the intervention respects the underlying physical constraints. This formulation ensures that all sampled interventions remain within the stage-specific feasible space while preserving diversity in intervention sequences under physical constraints. By using the differentiable mapping, we ensure that the intervention process is fully compatible with backpropagation, thus allowing the policy model to be trained end-to-end through gradient-based optimization.

## 3.3. Transition Model: Evolution of Lithography States

The transition model is designed to capture how process interventions drive the continuous evolution of lithography states within and across stages under physical constraints. Starting from the current state, the model predicts the next state by jointly conditioning on the process-window parameters and specific interventions, thereby supporting stable multi-step state extrapolation and closed-loop planning.

At evolution step $t$ of stage $s$, $\mathbf{z}_t^{(s)} \in \mathbb{R}^d$ and $\mathbf{a}_t^{(s)} \in \mathcal{S}^{(s)}$ denote the latent state and process intervention, respectively. The transition model $f_\phi(\cdot)$ employs the Diffusion Transformer (DiT) as the backbone to model the main state transition process and output a shared intermediate representation

for subsequent state updates. As lithography stages differ substantially in representation and value range, we employ stage-aware MLP heads after the DiT backbone. These select the corresponding mapping function based on the current stage to obtain the next state. The transition model can be represented as:

$$\mathbf{h}_{t+1}^{(s)} = \mathrm{DiT}_\phi\Big(\mathbf{z}_t^{(s)}, \mathbf{a}_t^{(s)}, \mathbf{p}\Big), \quad \mathbf{z}_{t+1}^{(s)} = \mathrm{MLP}_\phi^{(s)}\Big(\mathbf{h}_{t+1}^{(s)}\Big). \tag{11}$$

During training, since paired stage targets are available, we use a target-guided state-gate to stabilize multi-step evolution learning. After generating the next state $\mathbf{z}_{t+1}^{(s)}$, we assess whether the evolution within the current stage has sufficiently progressed toward the stage target observed in data. Let $\mathbf{x}^{*(s)}$ denote the ground-truth state at stage $s$ and $\mathbf{z}^{*(s)} = E(\mathbf{x}^{*(s)})$ be its latent representation. We define a stage-alignment loss:

$$\mathcal{L}_{\mathrm{evo}}^{(t)} = \left\| \mathbf{z}_{t+1}^{(s)} - \mathbf{z}^{*(s)} \right\|_2^2. \tag{12}$$

We compute its gradient norm with respect to interventions:

$$g_t = \left\| \nabla_{\mathbf{a}_t^{(s)}} \mathcal{L}_{\mathrm{evo}}^{(t)} \right\|_2. \tag{13}$$

where $\nabla_{\mathbf{a}_t^{(s)}}$ denotes the gradient with respect to $\mathbf{a}_t^{(s)}$. When $g_t > \tau_\nabla$, the current intervention has not yet sufficiently driven the state evolution into a locally stable regime, and the predicted state is fed back to the policy model within the same stage to continue intervention updates and state prediction. When $g_t \leq \tau_\nabla$, the evolution is considered locally stable, and we accept $\mathbf{z}_{t+1}^{(s)}$ as the stage output and proceed to the next stage. In practice, we set $\tau_\nabla = 5 \times 10^{-3}$ and cap the number of within-stage evolution iterations at 10. This criterion requires no additional training objectives.

## 3.4. Contrastive Variational Optimization Paradigm

In practical lithography datasets, only terminal states of each stage are observable, while stage-internal process interventions and intermediate states are unobserved. This renders explicit supervision of interventions infeasible. To address this, we propose the contrastive variational optimization paradigm that relies solely on terminal-stage supervision to jointly optimize process intervention distributions and state evolution, without requiring intermediate supervision.

### 3.4.1. VARIATIONAL INFERENCE

For a given initial condition and terminal outcome, there typically exist multiple physically admissible intervention sequences that can produce similar results. It makes direct inference of a single deterministic intervention ill-posed. Therefore, we formulate process intervention learning as an implicit variational inference problem, where intervention

sequences are treated as latent variables, and their posterior is inferred from terminal-state supervision. Specifically, the policy model $\pi_\theta$ parameterizes a tractable family of conditional distributions to approximate the implicit intervention posterior induced by terminal supervision and physical feasibility constraints. From a variational perspective, $\pi_\theta$ serves as a learnable approximation to this posterior. We define the stage-wise variational objective as:

$$\mathcal{L}_{\mathrm{var}}^{(s)} = \mathbb{E}_{\{\mathbf{c}_t^{(s)}\}_{t=0}^{T-1}} \left[ \mathcal{L}_{\mathrm{rec}}^{(s)} \Big( D(\mathbf{z}_T^{(s)}), \mathbf{x}^{*(s)} \Big) \right],$$
$$\mathbf{c}_t^{(s)} \sim \pi_\theta \left( \cdot \mid \mathbf{z}_t^{(s)}, \mathbf{p} \right), \quad \mathbf{a}_t^{(s)} = \mathbf{B}^{(s)} \mathbf{c}_t^{(s)}, \tag{14}$$

where $\mathbf{z}_T^{(s)}$ is obtained by rolling out the transition model for $T$ steps using the sampled interventions $\{\mathbf{a}_t^{(s)}\}_{t=0}^{T-1}$, and $D(\cdot)$ denotes the decoder. Minimizing $\mathcal{L}_{\mathrm{var}}^{(s)}$ serves as a variational surrogate that concentrates probability mass on intervention sequences that best explain the observed terminal state under the physical feasibility constraints. Unlike ELBO-based formulations that require an explicit likelihood, the intervention posterior here is implicitly specified by terminal-state supervision and the physics-informed intervention space, enabling stochastic yet physically consistent intervention inference without stage-internal supervision.

### 3.4.2. CONTRASTIVE LEARNING

To enforce stage-consistent interventions, we construct contrastive pairs by projecting the same coefficients onto physics-informed spaces of different stages. Given a sampled coefficient vector $\mathbf{c}_t^{(s)} \sim \pi_\theta(\cdot \mid \mathbf{z}_t^{(s)}, \mathbf{p})$, we form a positive (stage-matched) intervention and a negative (stage-mismatched) intervention:

$$\mathbf{a}_{t,\mathrm{pos}}^{(s)} = \mathbf{B}^{(s)} \mathbf{c}_t^{(s)}, \quad \mathbf{a}_{t,\mathrm{neg}}^{(s)} = \mathbf{B}^{(s+1)} \mathbf{c}_t^{(s)}. \tag{15}$$

Rolling out one step with the same transition model yields:

$$\mathbf{z}_{t+1,\mathrm{pos}}^{(s)} = f_\phi\Big(\mathbf{z}_t^{(s)}, \mathbf{a}_{t,\mathrm{pos}}^{(s)}, \mathbf{p}\Big),$$
$$\mathbf{z}_{t+1,\mathrm{neg}}^{(s)} = f_\phi\Big(\mathbf{z}_t^{(s)}, \mathbf{a}_{t,\mathrm{neg}}^{(s)}, \mathbf{p}\Big). \tag{16}$$

Using the stage $s$ terminal observation $\mathbf{x}^{*(s)}$ as supervision, we define a margin-based contrastive loss:

$$\mathcal{L}_{\mathrm{ctr}}^{(s)} = \max\Big(0, \ m + \mathcal{L}_{\mathrm{rec}}^{(s)}\big(\mathbf{x}_{t+1,\mathrm{pos}}^{(s)}, \mathbf{x}^{*(s)}\big)$$
$$- \mathcal{L}_{\mathrm{rec}}^{(s)}\big(\mathbf{x}_{t+1,\mathrm{neg}}^{(s)}, \mathbf{x}^{*(s)}\big)\Big). \tag{17}$$

where $\mathbf{x}_i^{(s)}$ is the image obtained from $\mathbf{z}_i^{(s)}$ by our Decoder, $m$ is set to 0.1. This objective pulls the stage-matched evolution toward $\mathbf{x}^{*(s)}$ while pushing the stage-mismatched evolution away, enforcing stage-specific consistency without requiring intermediate-state supervision.

*Table 1.* Evaluate the final evolution quality of mask, resist image, and ADI on the ID lithography dataset. The values before and after "/" denote the results of the forward evolution and inverse planning tasks, respectively. **Best** and second-best values are highlighted.

| Task | Method | Multi-Step | Multi-Stage | mPA(%)↑ | mIoU(%)↑ | F1(%)↑ | MSE($\times 10^{-3}$)↓ | EPE$_{avg}$(nm)↓ |
|---|---|---|---|---|---|---|---|---|
| Mask Evolution | LithoNet (Shao et al., 2020) | × | × | 93.54 / - | 85.70 / - | 84.29 / - | 57.92 / - | 46.16 / - |
| | LMLitho (Wang et al., 2025) | × | × | 96.27 / - | 91.95 / - | 95.99 / - | 19.03 / - | 7.12 / - |
| | Unitho (Jin et al., 2025a) | × | ✓ | 95.71 / - | 92.24 / - | 96.42 / - | 25.40 / - | 8.69 / - |
| | DINO-WM (Zhou et al., 2024) | ✓ | × | 93.47 / 94.83 | 90.04 / 92.88 | 93.00 / 93.46 | 26.51 / 25.91 | 21.96 / 18.33 |
| | DriveDreamer-2 (Zhao et al., 2025) | ✓ | × | 93.65 / 95.04 | 83.17 / 86.51 | 90.76 / 92.43 | 63.48 / 62.17 | 4.94 / 4.53 |
| | NWM (Bar et al., 2025) | ✓ | × | 91.81 / 92.77 | 85.95 / 87.62 | 90.75 / 92.22 | 29.65 / 28.07 | 26.36 / 23.90 |
| | **LithoDreamer (ours)** | ✓ | ✓ | **98.69 / 99.04** | **96.91 / 97.61** | **99.66 / 99.69** | **13.74 / 13.07** | **1.58 / 1.32** |
| Resist Image Evolution | LithoNet (Shao et al., 2020) | × | × | 91.41 / - | 82.14 / - | 74.36 / - | 78.67 / - | 33.31 / - |
| | LMLitho (Wang et al., 2025) | × | × | 95.23 / - | 96.35 / - | 96.24 / - | 18.66 / - | 6.31 / - |
| | Unitho (Jin et al., 2025a) | × | ✓ | 95.52 / - | 91.98 / - | 96.02 / - | 25.16 / - | 8.62 / - |
| | DINO-WM (Zhou et al., 2024) | ✓ | × | 95.68 / 96.32 | 93.89 / 94.72 | 95.03 / 95.83 | 22.47 / 20.39 | 17.37 / 15.03 |
| | DriveDreamer-2 (Zhao et al., 2025) | ✓ | × | 92.59 / 93.75 | 78.74 / 79.21 | 87.21 / 88.82 | 74.06 / 71.50 | 7.72 / 6.97 |
| | NWM (Bar et al., 2025) | ✓ | × | 91.81 / 92.58 | 85.95 / 86.89 | 90.75 / 91.79 | 55.43 / 53.45 | 26.36 / 24.16 |
| | **LithoDreamer (ours)** | ✓ | ✓ | **99.01 / 99.34** | **97.77 / 98.35** | **99.73 / 99.61** | **10.93 / 10.01** | **0.96 / 0.89** |
| ADI Evolution | LithoNet (Shao et al., 2020) | × | × | 61.17 / - | 8.91 / - | 26.78 / - | 247.82 / - | 13.39 / - |
| | LMLitho (Wang et al., 2025) | × | × | 52.56 / - | 40.96 / - | 32.14 / - | 162.41 / - | 5.54 / - |
| | Unitho (Jin et al., 2025a) | × | ✓ | - / - | - / - | - / - | - / - | - / - |
| | DINO-WM (Zhou et al., 2024) | ✓ | × | 73.57 / 74.69 | 69.41 / 71.83 | 69.99 / 71.12 | 120.51 / 113.98 | 19.71 / 19.42 |
| | DriveDreamer-2 (Zhao et al., 2025) | ✓ | × | 74.93 / 75.73 | 14.58 / 56.78 | 24.74 / 26.48 | 30.48 / 28.12 | 5.58 / 5.06 |
| | NWM (Bar et al., 2025) | ✓ | × | 71.43 / 73.27 | 61.35 / 69.72 | 53.20 / 54.83 | 137.32 / 132.40 | 26.36 / 25.29 |
| | **LithoDreamer (ours)** | ✓ | ✓ | **82.56 / 84.39** | **78.27 / 82.97** | **80.91 / 83.61** | **30.29 / 26.75** | **3.74 / 3.28** |

# 4. Experiments

Experiments have two tasks: forward evolution and inverse planning. Forward evolution: from layouts and process parameters, explore interventions to generate Mask, Resist Image, and ADI, assessing state evolution. Inverse planning: given target ADIs (28nm data (He et al., 2026), given target Resist Images), optimize interventions to drive state evolution toward targets, testing goal-directed planning.

## 4.1. Experimental Settings

### 4.1.1. IMPLEMENTATION DETAILS

We use a Variational Autoencoder (VAE) (Pu et al., 2016) and BERT (Devlin et al., 2019) as the image and text encoders, respectively. The policy model adopts vit-base-patch16-224 (Dosovitskiy, 2020), while the transition model uses DiT-B/2 (Peebles & Xie, 2023). The model is trained in an end-to-end pipeline with ground-truth inputs at each stage to mitigate error accumulation. The objective combines the variational loss Equation (14) and contrastive loss Equation (17), where the variational loss includes a diversified reconstruction loss comprising Mean Squared Error (MSE) loss, Binary Cross-Entropy (BCE) loss, Dice loss, and Edge loss. We use AdamW with an initial learning rate of 1e-6, weight decay of 0.01, $\beta_1 = 0.9$, and $\beta_2 = 0.999$. Training is conducted on 8 NVIDIA A100 GPUs for 10 epochs with a batch size of 8 and a linear learning-rate decay schedule.

### 4.1.2. DATASETS

As described in Section 2.1, we construct a large-scale 55 nm dataset covering the complete "Layout-Mask-Resist Image-ADI" pipeline. The dataset considers four process

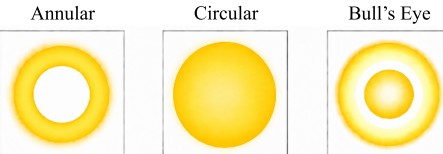

*Figure 3.* Three types of light sources in the dataset, each with configurable parameters, such as radius.

parameters: source type (Figure 3: Annular, Circular, Bull's Eye), resist threshold (0.09231251, 0.1236402, 0.1436665), focus (0 nm, 50 nm), and exposure dose (1.0×, 1.2×), resulting in 36 process configurations. The training set contains 280k samples, and the ID test set contains 20k samples, covering all configurations. We also construct two OOD test sets to evaluate generalization under unseen process parameters and different technology nodes. The first contains 3k samples from an unseen process setting at the same 55 nm node, namely Annular source, threshold 0.119340, focus 0 nm, and dose 1.0×. The second uses the public 28 nm LithoSim dataset (He et al., 2026) for cross-node evaluation.

### 4.1.3. EVALUATION METRICS

We evaluate lithography tasks under ID and OOD settings using five metrics. Mean Pixel Accuracy (mPA) and Mean Intersection over Union (mIoU) assess pixel-level accuracy and region consistency, respectively. Edge F1-Score (F1) evaluates boundary quality by balancing edge precision and recall. Mean Squared Error (MSE) measures overall intensity deviation. Edge Placement Error (EPE) is a manufacturing-critical contour-level metric in lithography. Following the gauge-based calculation protocol in Appendix B, we first convert the target and predicted pat-

*Table 2.* Comparison of Lithography Rule Check (LRC) violations between generated resist images and commercial simulation results on the ID dataset under the forward evolution task. The commercial tool is used as the reference; **best** and second-best values indicate the smallest and second-smallest absolute deviations among learned models (the process parameter is defined as Source_Threshold_Focus_Dose).

| Process Conditions | The Commercial Tool | | | LMLitho (Wang et al., 2025) | | | Unitho (Jin et al., 2025a) | | | **LithoDreamer (ours)** | | |
|---|---|---|---|---|---|---|---|---|---|---|---|---|
| | #Pinch | #Bridge | #EPE | #Pinch | #Bridge | #EPE | #Pinch | #Bridge | #EPE | #Pinch | #Bridge | #EPE |
| A[+] | 1289 | 2784 | 12732 | 1202(-6.7%) | 2731(-1.9%) | 12509(-1.8%) | 1219(-5.4%) | 2629(-5.6%) | 13009(2.2%) | 1217(-5.6%) | 2724(-2.2%) | 12920(1.5%) |
| B[+] | 1335 | 3032 | 12134 | 1292(-3.2%) | 2994(-1.3%) | 11366(-6.3%) | 1279(-4.2%) | 2866(-5.5%) | 13729(13.1%) | 1334(-0.1%) | 3095(2.1%) | 13160(8.5%) |
| C[+] | 1443 | 2657 | 5980 | 1584(9.8%) | 1765(-33.6%) | 7009(17.2%) | 2586(79.2%) | 3767(41.8%) | 12104(102.4%) | 1577(9.3%) | 2683(1.0%) | 5427(-9.2%) |
| D[+] | 1441 | 2795 | 6096 | 2819(95.6%) | 2811(0.6%) | 11489(88.5%) | 2808(94.9%) | 2965(6.1%) | 8764(43.8%) | 1407(-2.4%) | 3849(37.7%) | 6621(8.6%) |
| E[+] | 1456 | 2808 | 7041 | 2438(67.4%) | 2471(-12.0%) | 9712(37.9%) | 2460(69.0%) | 2910(3.6%) | 8650(22.9%) | 1519(4.3%) | 2761(-1.7%) | 6037(-14.3%) |
| Average | 1393 | 2815 | 8797 | 1867(34.0%) | 2554(-9.3%) | 10417(18.4%) | 2070(48.6%) | 3027(7.5%) | 11251(27.9%) | 1411(1.3%) | 3022(7.4%) | 8833(0.4%) |

[+] A: Bull's Eye_0.09231251_0 nm_1.2×; B: Bull's Eye_0.1436665_0 nm_1.2×; C: Annular_0.09231251_50 nm_1.0×;
D: Circular_0.1436665_0 nm_1.0×; E: Circular_0.09231251_0 nm_1.0×.

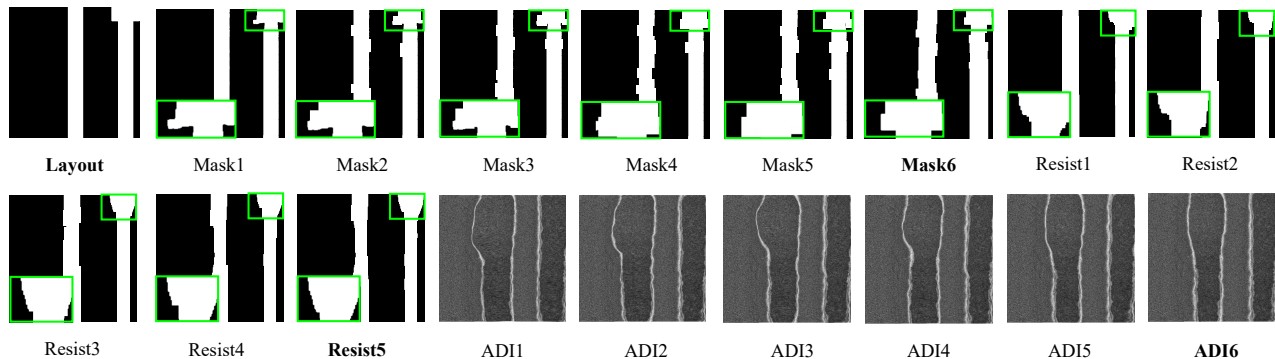

**Layout**  Mask1  Mask2  Mask3  Mask4  Mask5  **Mask6**  Resist1  Resist2

Resist3  Resist4  **Resist5**  ADI1  ADI2  ADI3  ADI4  ADI5  **ADI6**

*Figure 4.* Visualization of inverse planning on the ID dataset. Given the input layout and target ADI, LithoDreamer plans latent interventions and evolves the Mask, Resist Image, and ADI state to achieve the target pattern.

terns into polygonal contours, sample gauge points along the target contour, and measure the local displacement from the target contour to the predicted contour along the target normal direction. This metric directly reflects nanoscale contour placement fidelity and is therefore more manufacturing-relevant than generic image similarity metrics.

### 4.2. Main Experiment Results

We compare with classical lithography models and general WMs. Unlike conventional lithography models that are evaluated independently at each stage with ground-truth state inputs, LithoDreamer performs continuous prediction based on generated states from previous stages, thereby facing more pronounced cross-stage error accumulation.

#### 4.2.1. IN-DOMAIN RESULTS

As shown in Table 1, LithoDreamer achieves the best ***forward evolution*** quality across Mask, Resist Image, and ADI on the ID dataset. It obtains EPE values of 1.58 nm, 0.96 nm, and 3.74 nm, respectively, substantially outperforming classical lithography models and general WMs (Bar et al., 2025), and reducing EPE to the 1-4 nm range. It also yields consistent gains in region and boundary metrics, especially for ADI, where mIoU and F1 increase to 78.27% and 80.91%. Unlike static predictors or stage-wise mapping

methods (*i.e.*, Unitho (Jin et al., 2025a)), LithoDreamer explicitly models the continuous physical evolution of the "Layout-Mask-Resist Image-ADI" pipeline, enabling stable multi-step reasoning within stages and coherent cross-stage prediction. Further LRC validation in Table 2 shows that the generated Resist Images closely match commercial simulation results in Pinch, Bridge, and EPE violations, with average deviations of only 1.3%, 7.4%, and 0.4%, respectively. Compared with LMLitho (Wang et al., 2025) and Unitho (Jin et al., 2025a), which show much larger average deviations in Pinch and EPE violations, LithoDreamer better preserves fine-grained geometry and manufacturing-rule consistency, supporting its reliability for downstream lithography analysis.

For the ***inverse planning*** task on the ID dataset, Litho-Dreamer shows strong goal-directed planning capability, achieving EPE values of 1.32 nm, 0.89 nm, and 3.28 nm on Mask, Resist Image, and ADI, respectively, while consistently outperforming WM baselines in mPA, mIoU, and F1. Furthermore, the evolution trajectory in Figure 4 shows that LithoDreamer does not simply fit the terminal image, but progressively corrects local contour deviations through the Mask and Resist Image stages, leading to continuous morphology convergence in ADI. These results demonstrate its ability to perform stable target-directed evolution through physically consistent intervention planning.

*Table 3.* Evaluate the final evolution quality of mask, resist image, and ADI on two OOD lithography datasets. The values before and after "/" denote the results of the forward evolution and inverse planning tasks, respectively. **Best** and second-best values are highlighted.

| Dataset | Method | Mask | | | Resist Image | | | ADI | | |
|---|---|---|---|---|---|---|---|---|---|---|
| | | mPA↑ | mIoU↑ | EPE$_{avg}$↓ | mPA↑ | mIoU↑ | EPE$_{avg}$↓ | mPA↑ | mIoU↑ | EPE$_{avg}$↓ |
| 55nm with F+ | LithoNet (Shao et al., 2020) | 91.43 / - | 55.95 / - | 8.19 / - | 89.55 / - | 46.67 / - | 8.64 / - | 72.24 / - | 6.07 / - | 13.45 / - |
| | LMLitho (Wang et al., 2025) | 90.69 / - | 66.58 / - | 28.00 / - | 93.29 / - | 66.44 / - | 7.00 / - | 47.45 / - | 42.46 / - | 75.61 / - |
| | Unitho (Jin et al., 2025a) | 91.01 / - | 67.61 / - | 10.54 / - | 94.80 / - | 71.84 / - | 1.93 / - | - / - | - / - | - / - |
| | DINO-WM (Zhou et al., 2024) | 89.87 / 90.57 | 75.98 / 78.05 | 15.67 / 14.81 | 93.51 / 93.97 | 76.25 / 78.44 | 2.69 / 2.31 | 71.81 / 73.42 | 64.06 / 64.59 | 4.28 / 4.13 |
| | DriveDreamer-2 (Zhao et al., 2025) | 88.44 / 91.17 | 54.72 / 63.00 | 6.26 / 5.72 | 94.27 / 94.91 | 78.21 / 80.30 | 5.22 / 4.92 | 68.11 / 70.04 | 9.04 / 50.49 | 5.49 / 5.12 |
| | NWM (Bar et al., 2025) | 84.04 / 86.21 | 75.92 / 78.09 | 26.36 / 22.73 | 88.25 / 89.61 | 80.29 / 82.53 | 3.04 / 2.94 | 70.94 / 72.43 | 60.96 / 66.57 | 4.62 / 4.39 |
| | **LithoDreamer (ours)** | **96.92 / 97.33** | **77.52 / 82.79** | **1.77 / 1.51** | **98.27 / 98.64** | **82.91 / 84.07** | **1.06 / 0.97** | **78.89 / 85.73** | **76.04 / 84.66** | **4.05 / 3.21** |
| 28nm | LithoNet (Shao et al., 2020) | 93.06 / - | 31.49 / - | 23.49 / - | 93.06 / - | 31.49 / - | 20.52 / - | - / - | - / - | - / - |
| | LMLitho (Wang et al., 2025) | 82.49 / - | 30.64 / - | 4.63 / - | 92.46 / - | 29.08 / - | 7.90 / - | - / - | - / - | - / - |
| | Unitho (Jin et al., 2025a) | 81.20 / - | 33.98 / - | 4.78 / - | 91.71 / - | 30.59 / - | 5.33 / - | - / - | - / - | - / - |
| | DINO-WM (Zhou et al., 2024) | 73.90 / 75.12 | 62.37 / 64.99 | 5.85 / 5.42 | 75.81 / 77.00 | 61.32 / 63.74 | 5.97 / 5.03 | - / - | - / - | - / - |
| | DriveDreamer-2 (Zhao et al., 2025) | 86.30 / 87.27 | 18.81 / 30.08 | 15.03 / 14.26 | 80.75 / 83.95 | 20.39 / 36.15 | 19.38 / 18.46 | - / - | - / - | - / - |
| | NWM (Bar et al., 2025) | 71.24 / 72.35 | 58.92 / 60.79 | 20.81 / 19.62 | 78.94 / 79.68 | 60.70 / 62.76 | 29.69 / 28.53 | - / - | - / - | - / - |
| | **LithoDreamer (ours)** | **94.92 / 95.74** | **89.75 / 92.63** | **1.82 / 1.73** | **96.99 / 97.68** | **81.16 / 82.99** | **1.37 / 1.28** | - / - | - / - | - / - |

+ F: Source_Threshold_Focus_Dose: Annular_0.119340_0 nm_1.0×. ADI data are not provided in the 28 nm dataset (He et al., 2026).

*Table 4.* Ablation of the SPA design.

| Method | Mask | | Resist Image | | ADI | |
|---|---|---|---|---|---|---|
| | mPA↑ | EPE$_{avg}$↓ | mPA↑ | EPE$_{avg}$↓ | mPA↑ | EPE$_{avg}$↓ |
| w/o SPA | 70.02 | 28.06 | 71.67 | 28.41 | 58.66 | 38.47 |
| Random $\mathcal{S}$ | 83.79 | 27.04 | 83.98 | 25.82 | 64.13 | 33.91 |
| Shared SPA | 86.17 | 24.53 | 86.92 | 18.14 | 67.26 | 26.47 |
| Global PCA | 88.01 | 18.74 | 90.37 | 9.10 | 70.58 | 16.46 |
| SPA ($k = 2$) | 92.17 | 6.92 | 94.56 | 2.31 | 76.22 | 10.91 |
| **SPA ($k = 8$)** | **98.69** | **1.58** | **99.01** | **0.96** | **82.56** | **3.74** |
| SPA ($k = 16$) | 96.42 | 2.96 | 97.68 | 1.78 | 80.03 | 6.59 |

### 4.2.2. OUT-OF-DOMAIN RESULTS

As shown in Table 3, under unseen process parameter configurations at the same 55 nm node, LithoDreamer demonstrates stable OOD generalization in the ***forward evolution*** task. Compared with the static predictor LMLitho (Wang et al., 2025), LithoDreamer reduces the EPE by 26.23 nm, 5.94 nm, and 71.56 nm on Mask, Resist Image, and ADI, respectively, suggesting that it does not memorize specific process configurations but learns physically constrained evolution directions for stable extrapolation within the same node. On the cross-node LithoSim dataset (He et al., 2026), LithoDreamer still achieves the highest mIoU and reduces the EPE by 4.03 nm and 4.60 nm on Mask and Resist Image compared with DINO-WM (Zhou et al., 2024). This shows that ours can capture the effective influence of process interventions on lithography state evolution across different nodes, enabling stable extrapolation to unseen nodes.

In the OOD ***inverse planning*** task, LithoDreamer further demonstrates robust target-driven generalization. For the unseen 55 nm process configuration, LithoDreamer achieves EPE values of 1.51 nm, 0.97 nm, and 3.21 nm on Mask, Resist Image, and ADI, respectively, while generally outperforming executable WM baselines in mPA and mIoU. On the cross-node 28 nm LithoSim dataset (He et al., 2026), LithoDreamer keeps the EPE of Mask and Resist Image

at 1.73 nm and 1.28 nm, respectively, while achieving the highest mIoU. These results show that, even under unseen process parameters and different technology nodes, LithoDreamer can drive state evolution toward the target through physically consistent latent intervention planning, rather than merely fitting patterns within the training distribution.

### 4.3. Ablation Results

We conduct ablation studies on the ***ID forward evolution*** task to validate the effectiveness of the proposed architecture and key optimization modules. **Best** values are highlighted.

#### 4.3.1. DESIGN OF THE SPA

Table 4 reports the ablation results of the proposed SPA method. Removing SPA increases the ADI EPE from 3.74 nm to 38.47 nm, showing that unconstrained intervention-driven evolution is unstable. Random latent spaces and global PCA provide only limited gains, indicating that generic dimensionality reduction cannot capture stage-specific evolution directions. Moreover, sharing a single SPA across stages underperforms the stage-specific SPA, confirming that different lithography stages are governed by distinct physical dynamics. Sensitivity analysis further shows that a moderate latent space dimension ($k = 8$) achieves the best trade-off between expressiveness and physical constraint. Overall, results validate SPA as a critical and physics-informed constraint rather than a simple regularization or PCA-based approximation.

#### 4.3.2. DESIGN OF THE POLICY MODEL

As shown in Table 5, the ablation study validates the policy model from three aspects: contrastive constraints, stochastic intervention modeling, and process-parameter conditioning. Removing contrastive learning increases the EPE by 4.52

*Table 5.* Ablation of policy model design.

| Method | Mask | | Resist Image | | ADI | |
|---|---|---|---|---|---|---|
| | mPA↑ | EPE$_{avg}$↓ | mPA↑ | EPE$_{avg}$↓ | mPA↑ | EPE$_{avg}$↓ |
| w/o Contrastive | 93.83 | 6.10 | 88.71 | 6.97 | 78.27 | 7.89 |
| $\mathbf{c}_t^{(s)} = \boldsymbol{\mu}_t^{(s)}$ | 86.73 | 15.37 | 84.57 | 20.98 | 70.53 | 21.33 |
| w/o $\boldsymbol{\sigma}_t^{(s)}$ | 87.29 | 14.01 | 87.54 | 15.86 | 74.36 | 17.24 |
| w/o $\mathbf{p}$ | 95.34 | 8.91 | 96.32 | 11.26 | 79.13 | 10.75 |
| **Ours** | **98.69** | **1.58** | **99.01** | **0.96** | **82.56** | **3.74** |

*Table 6.* Ablation of our transition model design.

| Method | Mask | | Resist Image | | ADI | |
|---|---|---|---|---|---|---|
| | mPA↑ | EPE$_{avg}$↓ | mPA↑ | EPE$_{avg}$↓ | mPA↑ | EPE$_{avg}$↓ |
| w/o $\mathbf{a}_t^{(s)}$ | 90.87 | 6.32 | 85.66 | 7.80 | 73.58 | 9.02 |
| Shared Head | 95.61 | 4.38 | 86.59 | 5.02 | 66.50 | 26.86 |
| **Ours** | **98.69** | **1.58** | **99.01** | **0.96** | **82.56** | **3.74** |

*Table 7.* Ablation of adaptive state transfer determination.

| Method | Mask | | | Resist Image | | | ADI | | |
|---|---|---|---|---|---|---|---|---|---|
| | mPA↑ | EPE$_{avg}$↓ | Step$_{avg}$ | mPA↑ | EPE$_{avg}$↓ | Step$_{avg}$ | mPA↑ | EPE$_{avg}$↓ | Step$_{avg}$ |
| step = 3 | 94.56 | 4.90 | 3 | 95.93 | 4.34 | 3 | 77.83 | 19.82 | 3 |
| step = 5 | 97.03 | 3.42 | 5 | 97.72 | 3.41 | 5 | 80.02 | 5.96 | 5 |
| w/o Gradient | 97.02 | 3.60 | 6 | 98.00 | 3.32 | 8 | 80.13 | 5.67 | 9 |
| **Ours** | **98.69** | **1.58** | 6 | **99.01** | **0.96** | 4 | **82.56** | **3.74** | 8 |

nm, 6.01 nm, and 4.13 nm on Mask, Resist Image, and ADI compared with the full model, indicating that terminal-state variational supervision alone cannot effectively distinguish stage-matched from stage-mismatched intervention directions, weakening stage-consistent boundary evolution.

Stochastic intervention modeling is also critical. When the policy collapses to a deterministic intervention, $\mathbf{c}_t^{(s)} = \boldsymbol{\mu}_t^{(s)}$, the EPE increases to 15.37 nm, 20.98 nm, and 21.33 nm across the three stages, showing that a single point estimate cannot capture the underdetermined intervention-state evolution relationship. Removing uncertainty modeling (w/o $\boldsymbol{\sigma}_t^{(s)}$) improves over the deterministic variant, reducing the EPE to 14.01 nm, 15.86 nm, and 17.24 nm across the three stages, but it remains far inferior to the full model. This suggests that the learned stochasticity is not merely noise injection, but a key mechanism for exploring effective intervention candidates within the physically feasible space. Moreover, removing the process parameter $\mathbf{p}$ still preserves a certain level of prediction ability, but the EPE remains much higher at 8.91 nm, 11.26 nm, and 10.75 nm, compared with 1.58 nm, 0.96 nm, and 3.74 nm for the full model. This demonstrates that explicit process conditioning is critical for stage-consistent intervention decisions. Overall, LithoDreamer achieves the highest mPA and lowest EPE across all stages, showing that these designs jointly improve intervention planning and state prediction.

### 4.3.3. Design of the Transition Model Architecture

As shown in Table 6, the ablation results highlight the importance of both intervention-aware transition modeling and stage-specific output mappings. Removing the process intervention $\mathbf{a}^{(s)}$ consistently degrades performance across all stages, increasing the EPE to 6.32 nm, 7.80 nm, and 9.02 nm for Mask, Resist Image, and ADI, respectively. This indicates that, without intervention-conditioned

inputs, the transition model cannot accurately capture the state evolution dynamics driven by process interventions. The Shared Head variant still maintains reasonable performance on Mask and Resist Image, but degrades substantially on ADI, with the EPE increasing to 26.86 nm and mPA dropping to 66.50%. This suggests that different lithography stages exhibit strong representational and distributional heterogeneity, making a single shared output head insufficient to handle binary masks, resist images, and ADI states simultaneously. LithoDreamer achieves the highest mPA and lowest EPE across all stages, validating the necessity of intervention-conditioned transition modeling and stage-aware output mappings for stable multi-stage evolution.

### 4.3.4. Design of Transfer State Determination

Table 7 evaluates inter-stage transfer and intra-stage iteration strategies of the transition model. Using a fixed number of iterations leads to a clear stage-dependent mismatch: with step = 3, all stages are under-evolved, and the degradation is most severe at ADI (EPE 19.82 nm), indicating that late-stage dynamics are more nonlinear and error-sensitive. Increasing the fixed budget to step = 5 substantially improves performance (ADI's EPE 5.96 nm), yet still underperforms adaptive iteration. And ours achieves the best results across all stages while allocating different average iteration counts, showing that different stages require different refinement depths, with ADI requiring the deepest refinement. Notably, w/o Gradient uses even more iterations on average (ADI 9) but yields inferior accuracy (ADI's EPE 5.67 nm vs 3.74 nm), demonstrating that the improvement does not stem from more iterations alone. These results confirm that gradient-based convergence criteria, rather than fixed or heuristic iteration budgets, are essential for stable inter-stage transfer and accurate intra-stage evolution.

## 5. Conclusion

We propose LithoDreamer, a physics-informed world model that formulates computational lithography as a decision-aware multi-stage physical evolution process. By learning stage-specific latent spaces with contrastive variational optimization, LithoDreamer jointly models process interventions and state transitions without intermediate supervision. Extensive ID and OOD experiments show superior accuracy and generalization, especially in edge placement precision.

## Acknowledgements

This work was supported by the Yangtze River Delta Science and Technology Innovation Community Joint Fundamental Research Project (Grant No. 2024CSJZN0500).

## Impact Statement

This paper presents work whose goal is to advance the field of machine learning. There are many potential societal consequences of our work, none of which we feel must be specifically highlighted here.

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

The appendix is divided into several sections, each providing additional experimental details, metric definitions, and qualitative analyses.

## A. Detailed Experimental Inference Setups

LithoDreamer is evaluated under two inference modes: autonomous forward evolution and target-conditioned inverse planning. In both modes, the encoder, decoder, policy model, transition model, and latent spaces are fixed at test time. Different from the target-guided state-gate criterion used during training, which relies on observed stage targets, inference adopts a target-free latent-convergence criterion to avoid using unavailable intermediate ground-truth states.

### A.1. Autonomous Forward Evolution

Given an input layout $\mathbf{x}^{(1)}$ and a process condition $\mathbf{p}$, forward evolution aims to autonomously explore process intervention sequences and evolve the lithography state through the multi-stage pipeline:

$$\text{Layout} \rightarrow \text{Mask} \rightarrow \text{Resist Image} \rightarrow \text{ADI}.$$

For each stage $s$, the current state at evolution step $t$ is encoded as $\mathbf{z}_t^{(s)} = E(\mathbf{x}_t^{(s)})$. The policy model generates stage-wise intervention coefficients conditioned on the current latent state and process condition:

$$\mathbf{c}_t^{(s)} \sim \pi_\theta(\cdot \mid \mathbf{z}_t^{(s)}, \mathbf{p}),$$

which are mapped to the stage-specific feasible space by:

$$\mathbf{a}_t^{(s)} = \mathbf{B}^{(s)} \mathbf{c}_t^{(s)}.$$

The transition model then predicts the next latent state:

$$\mathbf{z}_{t+1}^{(s)} = f_\phi(\mathbf{z}_t^{(s)}, \mathbf{a}_t^{(s)}, \mathbf{p}).$$

Within each stage, the evolution continues until the relative latent-state change satisfies:

$$r_t^{(s)} = \frac{\left\| \mathbf{z}_{t+1}^{(s)} - \mathbf{z}_t^{(s)} \right\|_2}{\left\| \mathbf{z}_t^{(s)} \right\|_2 + \delta} < \epsilon_{\text{lat}},$$

or the maximum number of evolution steps $K_s$ is reached. We set $\epsilon_{\text{lat}} = 5 \times 10^{-3}$, $K_s = 10$, and $\delta = 10^{-8}$ for numerical stability. The converged latent state is decoded as the output of the current stage and passed to the next stage. In this mode, process interventions are explored and generated online by the learned policy and are not optimized by test-time backpropagation.

### A.2. Target-Conditioned Inverse Planning

Given an input layout $\mathbf{x}^{(1)}$, a process condition $\mathbf{p}$, and a target ADI $\mathbf{x}_{\text{tar}}^{(\text{ADI})}$, inverse planning optimizes a feasible multi-stage intervention sequence to drive the final state toward the target (*in the 28 nm dataset (He et al., 2026), the given target is the Resist Images, and the subsequent formulas are based on the given target ADIs*). All network parameters and latent spaces remain frozen. The optimization variables are the stage-wise intervention coefficients $\{\mathbf{c}_t^{(s)}\}$, initialized from the explored policy prior. At each outer planning iteration, the coefficients are mapped into feasible interventions:

$$\mathbf{a}_t^{(s)} = \mathbf{B}^{(s)}\mathbf{c}_t^{(s)},$$

and a differentiable multi-stage evolution is performed using the same target-free stopping rule as forward evolution. After decoding the final ADI prediction $\hat{\mathbf{x}}^{(\text{ADI})}$, we optimize the intervention coefficients with the terminal planning objective:

$$\mathcal{L}_{\text{plan}} = \lambda_{\text{rec}}\mathcal{L}_{\text{rec}}\left(\hat{\mathbf{x}}^{(\text{ADI})}, \mathbf{x}_{\text{tar}}^{(\text{ADI})}\right) + \lambda_{\text{edge}}\mathcal{L}_{\text{edge}}\left(\hat{\mathbf{x}}^{(\text{ADI})}, \mathbf{x}_{\text{tar}}^{(\text{ADI})}\right) + \lambda_{\text{prior}}\mathcal{L}_{\text{prior}} + \lambda_{\text{smooth}}\mathcal{L}_{\text{smooth}},$$

where we set $\lambda_{\text{rec}} = 0.5$, $\lambda_{\text{edge}} = 0.5$, $\lambda_{\text{prior}} = 10^{-3}$, and $\lambda_{\text{smooth}} = 10^{-3}$. Here, $\mathcal{L}_{\text{edge}}$ denotes a differentiable edge-alignment loss used for inverse-planning optimization. In our implementation, it is computed from Sobel edge maps:

$$\mathcal{L}_{\text{edge}} = \left\|\mathcal{G}(\hat{\mathbf{x}}^{(\text{ADI})}) - \mathcal{G}(\mathbf{x}_{\text{tar}}^{(\text{ADI})})\right\|_1,$$

where $\mathcal{G}(\cdot)$ denotes the differentiable Sobel edge-magnitude operator. This loss encourages contour alignment during optimization, while the official EPE values reported in all experiments are computed using the gauge-based EPE in Appendix B. And the prior and smoothness terms are defined as:

$$\mathcal{L}_{\text{prior}} = \sum_{s,t}\left\|\mathbf{c}_t^{(s)} - \boldsymbol{\mu}_t^{(s)}\right\|_2^2, \quad \mathcal{L}_{\text{smooth}} = \sum_{s,t}\left\|\mathbf{c}_t^{(s)} - \mathbf{c}_{t-1}^{(s)}\right\|_2^2.$$

The planning objective is backpropagated only to $\{\mathbf{c}_t^{(s)}\}$, while all model parameters remain fixed. Inverse planning does not require a target Mask or target Resist Image; the optimization is driven only by the target ADI, and physical feasibility is enforced by the stage-specific latent spaces $\mathcal{S}^{(s)}$.

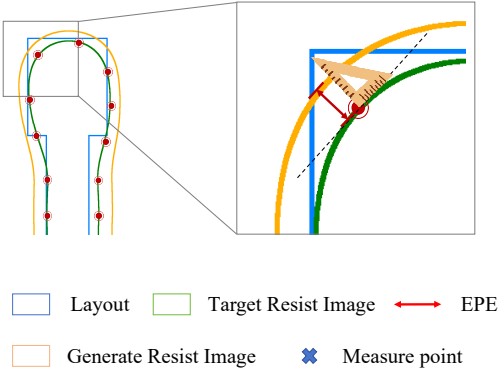

Layout   Target Resist Image   ⟷ EPE

Generate Resist Image   ✖ Measure point

*Figure 5.* Schematic illustration of gauge-based EPE measurement. Local measurement gauges are placed on the target resist image contour, and edge displacement is evaluated along the corresponding contour-normal direction. The magnified view highlights how the measured offset captures local contour placement deviation between the generated and target resist image patterns.

## B. Principles of Lithography Metric Calculation

Lithography evaluation cannot be fully characterized by generic image-similarity metrics, because nanometer-scale contour shifts and local topological failures may directly affect manufacturability. Therefore, in addition to pixel- and region-level

metrics, we report ***EPE*** and ***LRC-style violation counts***. All geometric distances are converted to nanometers using a fixed pixel-to-length scaling factor $\alpha$ shared across all methods and process conditions.

### B.1. Gauge-based EPE

As illustrated in Figure 5, EPE evaluates the local contour placement error between the target pattern and the generated pattern. The blue layout is shown as the design reference, while the green and orange contours denote the target resist image and generated resist image, respectively. EPE is measured by placing gauge points on the target contour and computing the displacement from the target contour to the generated contour along the local normal direction.

Let $I^{\mathrm{gt}}$ and $I^{\mathrm{pred}}$ denote the ground-truth and predicted patterns. We convert them into polygonal contours, yielding the target contour $\partial T$ and the predicted contour $\partial C$. Gauge points are uniformly sampled on $\partial T$:

$$V = \{p_i\}_{i=1}^{N}.$$

For each gauge point $p_i$, let $\mathbf{n}(p_i)$ be the outward unit normal of the target contour. The corresponding normal sampling line is defined as:

$$\ell(p_i) = \{p_i + t\mathbf{n}(p_i) \mid t \in \mathbb{R}\}.$$

The matched point on the predicted contour is selected as the closest intersection between $\ell(p_i)$ and $\partial C$:

$$q_i = \arg \min_{q \in \partial C \cap \ell(p_i)} \|q - p_i\|_2.$$

If no valid intersection exists, the gauge point is excluded from aggregation. We denote the remaining valid gauge set as: $V_{\mathrm{val}}$.

The signed local displacement at $p_i$ is computed as:

$$d(p_i) = \langle q_i - p_i, \mathbf{n}(p_i) \rangle,$$

where positive and negative values indicate outward and inward contour bias, respectively. The signed local EPE in physical units is:

$$\widetilde{\mathrm{EPE}}(p_i) = \alpha d(p_i).$$

The final reported EPE is the mean absolute local displacement:

$$\mathrm{EPE}_{\mathrm{avg}} = \frac{1}{|V_{\mathrm{val}}|} \sum_{p_i \in V_{\mathrm{val}}} \left| \widetilde{\mathrm{EPE}}(p_i) \right|.$$

This gauge-based protocol is used as the official EPE calculation for all reported results. The differentiable $\mathcal{L}_{\mathrm{edge}}$ used in inverse planning is a Sobel-based edge-alignment loss that encourages contour consistency during optimization. It is not used as our EPE metric; all reported EPE values follow the gauge-based protocol defined here.

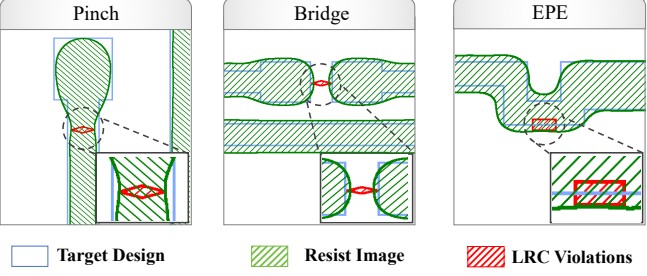

*Figure 6.* Representative LRC violation categories used for manufacturability assessment. Pinch captures locally narrowed printed features, Bridge captures unintended connections or insufficient spacing between neighboring structures, and EPE captures excessive contour displacement beyond the allowed placement tolerance. Red markers indicate the detected violation regions.

**B.2. LRC-style Violation Counts**

Beyond continuous EPE statistics, we further report LRC-style violation counts to assess manufacturing-rule consistency. As shown in Figure 6, we consider three representative lithography rule violations: Pinch, Bridge, and EPE. The blue contours denote the target design, the green hatched regions denote the generated resist image, and the red hatched regions indicate detected LRC violation markers.

A Pinch violation occurs when the local printed linewidth becomes smaller than the minimum allowable width. A Bridge violation occurs when two structures that should remain separated become unintentionally connected, or when their spacing falls below the allowed minimum. An EPE violation occurs when the magnitude of the signed local EPE exceeds the placement tolerance:

$$\left|\widetilde{\mathrm{EPE}}(p_i)\right| > \tau_{\mathrm{EPE}}, \quad p_i \in V_{\mathrm{val}}.$$

The verification flow returns violation markers for each category, and we report the corresponding marker counts:

$$N_{\mathrm{pinch}} = |\mathcal{M}_{\mathrm{pinch}}|, \quad N_{\mathrm{bridge}} = |\mathcal{M}_{\mathrm{bridge}}|, \quad N_{\mathrm{epe}} = |\mathcal{M}_{\mathrm{epe}}|.$$

All methods are evaluated using the same contour extraction, gauge sampling, verification deck, and marker reporting settings, ensuring that the reported EPE and LRC counts reflect lithography contour fidelity and manufacturability rather than measurement-configuration differences.

## C. Detailed Light Source

The illumination source is a key optical degree of freedom in computational lithography. It defines the angular distribution of light incident on the mask and determines how different diffraction orders and spatial-frequency components are transferred through the projection optics. Therefore, changing the source shape can affect aerial-image contrast, edge sharpness, process-window behavior, and the dominant failure modes of the printed pattern.

In our evaluation, we consider three representative source families, as shown in Figure 3: *Circular*, *Annular*, and *Bull's Eye*. These sources cover three typical illumination regimes: on-axis illumination, off-axis illumination, and mixed illumination.

**Circular source.** The circular source represents conventional on-axis illumination, where the illumination energy is distributed around the optical axis. This source provides relatively balanced imaging behavior and is effective for transferring low- and medium-spatial-frequency components. It is commonly suitable for isolated features or layouts without strong periodicity. However, for dense patterns or high-frequency structures, the limited angular diversity of circular illumination may reduce image contrast and increase sensitivity to edge-placement errors.

**Annular source.** The annular source concentrates illumination energy in an off-axis ring while suppressing the central illumination region. This off-axis configuration enhances the transfer of selected higher spatial-frequency components and is often beneficial for dense or periodic patterns. By changing the balance of diffraction orders captured by the projection optics, annular illumination can improve local image contrast and depth-of-focus behavior. Meanwhile, it may also introduce source-dependent contour shifts, line-end distortions, or different pinch/bridge tendencies compared with on-axis illumination.

**Bull's Eye source.** The Bull's Eye source combines a central circular component with an outer annular component. It therefore represents a mixed illumination condition that includes both on-axis and off-axis contributions. The central component helps preserve low-frequency information and isolated-feature fidelity, while the annular component enhances the transfer of higher-frequency information in dense regions. As a result, the Bull's Eye source provides a more balanced illumination regime and can produce contour behaviors distinct from either purely circular or purely annular illumination.

Together, these three source families provide representative coverage of illumination conditions commonly used in computational lithography. Since the source distribution directly changes the optical transfer characteristics, different source types can lead to different mask corrections, resist-image evolution, final contour morphology, and manufacturing-rule violations. Including these source families allows us to evaluate whether the proposed model can generalize across source-dependent lithographic variations rather than being limited to a single fixed imaging condition.

*Table 8.* Ablation of the number of sampled adjacent states for SPA basis estimation in the forward evolution task of the ID dataset. **Best** values are highlighted.

| Number | Mask | | | Resist Image | | | ADI | | |
|---|---|---|---|---|---|---|---|---|---|
| | mPA↑ | mIoU↑ | EPE$_{avg}$↓ | mPA↑ | mIoU↑ | EPE$_{avg}$↓ | mPA↑ | mIoU↑ | EPE$_{avg}$↓ |
| 1k | 90.17 | 86.76 | 3.47 | 92.78 | 87.64 | 2.73 | 70.35 | 67.18 | 8.07 |
| 5k | 95.01 | 89.38 | 2.02 | 96.26 | 95.06 | 7.41 | 78.93 | 74.36 | 5.61 |
| **10k** | **98.69** | **96.91** | **1.58** | **99.01** | **97.77** | **0.96** | **82.56** | **78.27** | **3.74** |
| 15k | 98.65 | 96.78 | 1.73 | 98.97 | 97.32 | 0.99 | 82.07 | 78.09 | 3.92 |
| 20k | 97.18 | 94.13 | 1.96 | 98.73 | 96.81 | 1.27 | 82.00 | 77.64 | 4.64 |

# D. Additional Experiments of Latent Spaces

This section provides additional experiments to further analyze the construction and transferability of the proposed stage-specific latent spaces. We first study how the number of adjacent state pairs affects SPA basis estimation, aiming to identify a suitable sample size for capturing physically meaningful evolution directions. We then validate whether the learned latent spaces can serve as transferable physical priors for general WMs by incorporating $\mathcal{S}^{(s)}$ into different world model baselines and evaluating their forward evolution and inverse planning performance.

## D.1. Design of Adjacent State Pairs in SPA

Table 8 evaluates the effect of the number of sampled adjacent states for SPA basis estimation on the ID forward evolution task. Increasing the sample size from 1k to 10k generally improves the final performance, although intermediate sample sizes may show stage-dependent fluctuations. In particular, the EPE is reduced by 1.89 nm, 1.77 nm, and 4.33 nm for Mask, Resist Image, and ADI, respectively, indicating that sufficient adjacent-state samples provide a more accurate estimate of physically admissible evolution directions.

However, further increasing the sample size beyond 10k does not bring additional benefits. Compared with 15k and 20k, the 10k setting maintains higher mPA and mIoU while achieving lower EPE, especially for ADI, where the EPE is lower than the 20k setting by 0.90 nm. This suggests that overly large sample sets may introduce non-local or heterogeneous variations, weakening the locality assumption of SPA. Overall, 10k offers the best balance between covering local state variations and preserving physically meaningful evolution directions.

## D.2. Validation of the Latent Space Constructed by SPA

As shown in Table 9, introducing the stage-specific latent spaces $\mathcal{S}^{(s)}$ consistently improves the forward evolution performance of general WMs under both ID and OOD settings. For DINO-WM (Zhou et al., 2024), $\mathcal{S}^{(s)}$ increases Mask mIoU by 3.00/6.59 points and reduces Mask EPE by 5.64/0.64 nm on ID/OOD data, showing stronger contour-level stability. For DriveDreamer-2 (Zhao et al., 2025), the gain is most pronounced in ADI, where mIoU improves by 41.43/30.58 points and mPA increases by 5.35/8.43 points, indicating that stage-specific latent spaces effectively mitigate cross-stage error accumulation. NWM also benefits substantially. These results demonstrate that $\mathcal{S}^{(s)}$ provides transferable physical constraints for generic WMs, leading to more coherent multi-stage lithography evolution.

Table 10 further shows that $\mathcal{S}^{(s)}$ improves goal-directed inverse planning. For DINO-WM (Zhou et al., 2024), adding $\mathcal{S}^{(s)}$ improves Mask mIoU by 2.74/8.72 points and reduces Mask EPE by 4.11/0.08 nm, suggesting more stable target-conditioned intervention search. These consistent gains indicate that stage-specific latent spaces constrain the inverse-planning trajectory toward physically feasible evolution directions, improving both target convergence and OOD robustness.

# E. Visual Comparisons of Forward Evolution

As shown in Figure 7, we compare different methods on an OOD sample at the 55 nm node. Although most baselines roughly preserve the input layout, their errors accumulate across stages: LithoNet (Shao et al., 2020) and LMLitho (Wang et al., 2025) introduce severe artifacts in Mask and ADI, Unitho (Jin et al., 2025a) produces distorted intermediate contours, and general WMs suffer from broken or unstable ADI morphologies. In contrast, LithoDreamer better maintains mask topology, resist-region continuity, and ADI contour fidelity, showing more coherent cross-stage evolution.

Figure 8 further evaluates LithoDreamer on curved and irregular OOD layouts. The results show that LithoDreamer can

*Table 9.* Forward evolution performance of general WMs with and without stage-specific latent spaces $\mathcal{S}^{(s)}$. Values before and after "/" denote ID/OOD (55nm with F) results. Results after incorporating $\mathcal{S}^{(s)}$ are **highlighted**.

| Model | Mask | | | Resist Image | | | ADI | | |
|---|---|---|---|---|---|---|---|---|---|
| | mPA↑ | mIoU↑ | EPE$_{avg}$↓ | mPA↑ | mIoU↑ | EPE$_{avg}$↓ | mPA↑ | mIoU↑ | EPE$_{avg}$↓ |
| DINO-WM (Zhou et al., 2024) | 93.47/89.87 | 90.04/75.98 | 21.96/15.67 | 95.68/93.51 | 93.89/76.25 | 17.37/2.96 | 73.57/71.81 | 69.41/64.06 | 19.71/4.28 |
| DINO-WM + $\mathcal{S}^{(s)}$ | **94.78/91.61** | **93.04/82.57** | **16.32/15.03** | **95.80/93.92** | **94.19/78.46** | **16.90/2.71** | **74.33/72.73** | **72.61/66.10** | **18.26/4.22** |
| DriveDreamer-2 (Zhao et al., 2025) | 93.65/88.44 | 83.17/54.72 | 4.94/6.26 | 92.59/94.27 | 78.74/78.21 | 7.72/5.22 | 74.93/68.11 | 14.58/9.04 | 5.58/5.49 |
| DriveDreamer-2 + $\mathcal{S}^{(s)}$ | **95.47/89.95** | **84.80/57.66** | **4.64/6.01** | **92.59/94.83** | **80.71/79.03** | **5.13/4.60** | **80.28/76.54** | **56.01/39.62** | **5.07/4.89** |
| NWM (Bar et al., 2025) | 91.81/84.04 | 85.95/75.92 | 26.36/26.36 | 91.81/88.25 | 85.95/80.29 | 26.36/3.04 | 71.43/70.94 | 61.35/60.96 | 26.36/4.62 |
| NWM + $\mathcal{S}^{(s)}$ | **93.94/88.62** | **90.74/83.67** | **23.81/24.07** | **94.72/93.65** | **89.20/87.90** | **22.51/2.93** | **79.66/76.51** | **72.04/70.83** | **19.91/4.02** |

*Table 10.* Target-conditioned inverse planning performance of general WMs with and without stage-specific latent spaces $\mathcal{S}^{(s)}$. Values before and after "/" denote ID/OOD (55nm with F) results. Results after incorporating $\mathcal{S}^{(s)}$ are **highlighted**.

| Model | Mask | | | Resist Image | | | ADI | | |
|---|---|---|---|---|---|---|---|---|---|
| | mPA↑ | mIoU↑ | EPE$_{avg}$↓ | mPA↑ | mIoU↑ | EPE$_{avg}$↓ | mPA↑ | mIoU↑ | EPE$_{avg}$↓ |
| DINO-WM (Zhou et al., 2024) | 94.83/90.57 | 92.88/78.05 | 18.33/14.81 | 96.32/93.97 | 94.72/78.44 | 15.03/2.31 | 74.69/73.42 | 71.83/64.95 | 19.42/4.13 |
| DINO-WM + $\mathcal{S}^{(s)}$ | **95.91/93.28** | **95.62/86.77** | **14.22/14.73** | **96.96/94.77** | **95.08/80.07** | **13.68/2.00** | **76.64/75.87** | **74.59/68.74** | **17.96/3.99** |
| DriveDreamer-2 (Zhao et al., 2025) | 95.04/91.17 | 86.51/63.00 | 4.53/5.72 | 93.75/94.91 | 79.21/80.30 | 6.97/4.92 | 75.73/70.04 | 56.78/50.49 | 5.06/5.12 |
| DriveDreamer-2 + $\mathcal{S}^{(s)}$ | **96.39/93.38** | **88.73/67.31** | **4.05/5.29** | **94.46/95.75** | **82.97/82.43** | **4.99/4.50** | **82.61/78.95** | **69.85/57.42** | **4.67/4.10** |
| NWM (Bar et al., 2025) | 92.77/86.21 | 87.62/78.09 | 23.90/22.73 | 92.58/89.61 | 86.89/82.53 | 24.16/2.94 | 73.27/72.43 | 69.72/66.57 | 25.29/4.39 |
| NWM + $\mathcal{S}^{(s)}$ | **94.57/89.95** | **92.67/87.02** | **21.98/20.36** | **94.90/93.67** | **90.03/88.74** | **20.79/2.69** | **80.06/78.97** | **75.81/72.66** | **19.53/3.87** |

preserve the main geometric correspondence from Layout to Mask and Resist Image, while capturing process-induced contour rounding in ADI. Figure 9 presents contact-like patterns with isolated features, showing that LithoDreamer can reproduce local shape deformation and ADI boundary evolution under unseen process conditions. Nevertheless, slight local contour deviations remain in some ADI predictions, suggesting that fine-scale boundary refinement under highly complex layouts can be further improved.

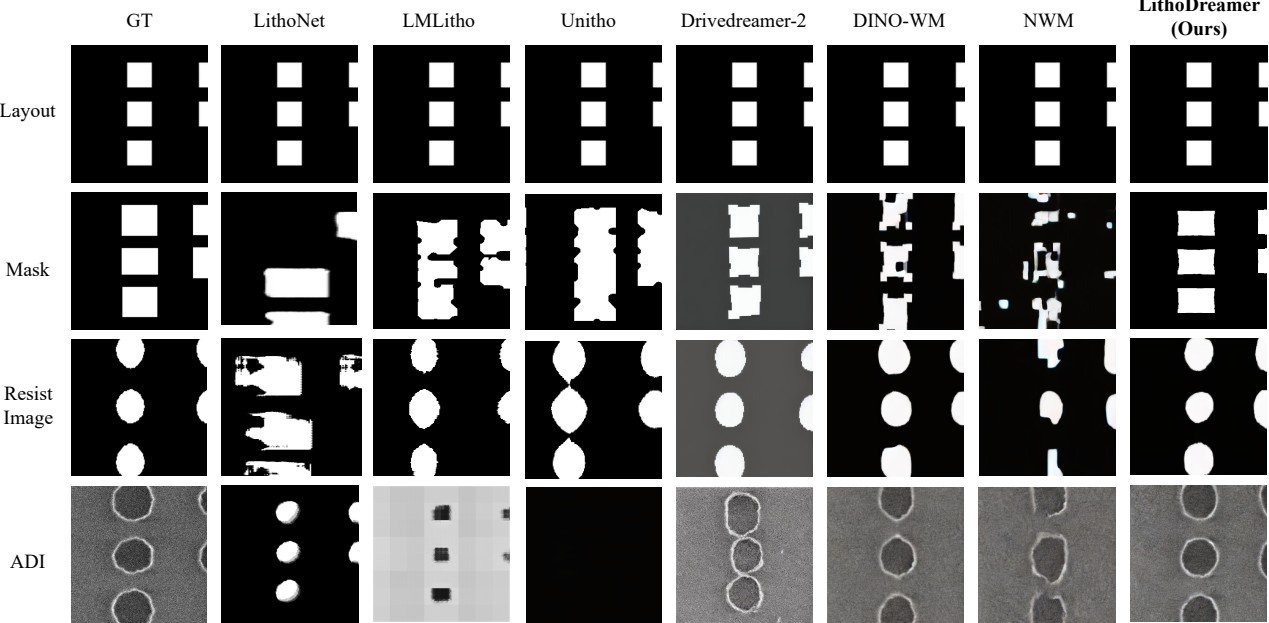

*Figure 7.* Qualitative comparison of forward evolution results on OOD samples at the 55 nm process node.

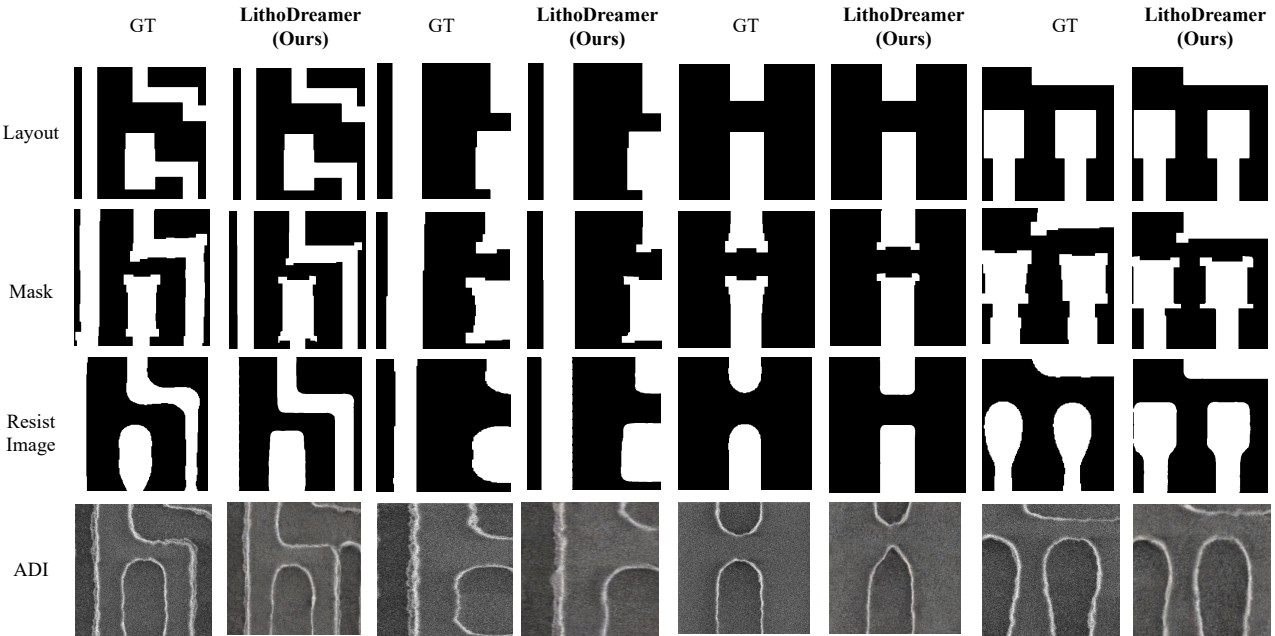

*Figure 8.* Forward evolution results on curved and irregular OOD layouts.

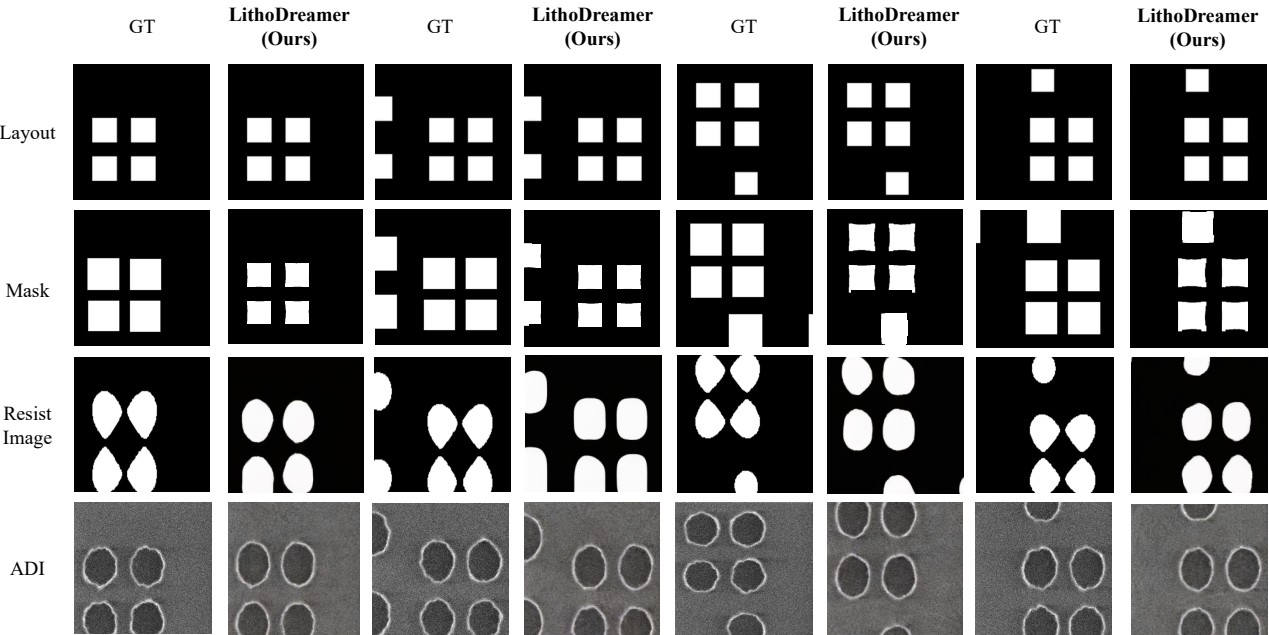

*Figure 9.* Forward evolution results on isolated contact-like OOD layouts.

