# OpenReview forum: "LithoDreamer: A Physics-Informed World Model for Multi-Stage Computational Lithography"
_ICML.cc/2026/Conference — ICML 2026 regular_

### Official Review · Reviewer_H19i · 2026-02-13

**Soundness:** 3
**Presentation:** 3
**Significance:** 3
**Originality:** 2
**Overall Recommendation:** 4
**Confidence:** 4

**Summary:**

This paper propose a framework for the computational lithography tasks, including 1) VAE for feature extractiona and image reconstruction, 2) state transformer blocks for dynamics modeling, 3) process transformer blocks for action decision. It achieves better performances among mask prediction, resist image prediction and ADI prediction tasks, compared to other Litho baselines and general-purpose world models.

**Compliance With Llm Reviewing Policy:**

Affirmed.

**Final Justification:**

In light of the authors' rebuttal and the subsequent discussions, I am raising my final recommendation to a Weak Accept. The authors' response directly addressed my primary concerns and positively changed my evaluation.

* **Originality and Significance:** While I still maintain that the individual algorithmic components (VAE, ViT, DiT) are standard, I acknowledge that the system-level integration is highly valuable. Formulating the `Layout → Mask → Resist Image → ADI` pipeline into a continuously evolving, intervention-aware system is a non-trivial and significant contribution to the computational lithography and EDA community.
* **Soundness:** My initial concerns regarding error propagation and the stretched "world model/physics-informed" framing were effectively resolved. The ablation data showing that adding explicit physical equations (EPE loss) actually increased OOD false-positives convincingly justifies their data-driven SPA approach. Furthermore, the newly added Target-Conditioned Inverse Planning experiment demonstrates practical multi-step inverse control, which makes the "World Model" framing much more acceptable.

**Key Questions For Authors:**

1. What is the "stage-specific physical dynamics" mentioned in the right column between line 154-155?
2. The authors claim that the proposed model is "physics-informed", could the authors give more detailed discussions about this statement? Only providing the inputs such as state x(s) is not enough to underpin this opinion. And why the latent space S(s) is physics-informed?

**Limitations:**

Limitations are not discussed. The input of the world models is focus on the computational lithography tasks, which is not simple sufficient to generalize to other chip design tasks which share the same or similar physical knowledge.

**Strengths And Weaknesses:**

Strengths
1. The paper is easy to follow and gives detailed calculation of each process.
2. Better performance on various tasks compared to other baselines.

Weaknesses
1. The "world model" framing is conceptually stretched. Standard WMs learn from real agent-environment interactions with observable actions, whereas here process interventions are latent variables inferred from data with no actual agent-environment loop. More critically, the WM baselines, including DriveDreamer-2, NWM and DINO-WM, are designed for video prediction and navigation, possessing no lithography domain knowledge whatsoever. The paper provides no detail on how these models were adapted for lithography tasks or whether they received comparable tuning effort.
2. The "physics-informed" claim is not substantiated. SPA (Eqs. 1–9) performs eigenvalue decomposition on the covariance matrix of stage-specific latent difference vectors, which is standard PCA applied to latent deltas. No physical equations or domain-specific physical constraints are incorporated into the formulation. The term physics-informed typically implies that known governing equations are embedded into the model structure or loss, yet SPA derives its basis purely from data statistics.
3. The framework assembles known components: a frozen VAE encoder, a ViT-based policy model, a DiT-based transition model, contrastive learning, and variational inference with reparameterization. Each piece is standard, and the combination, while reasonable as engineering, does not introduce a novel algorithmic or architectural contribution. The value lies more in the domain application than in methodological innovation.
4. The training procedure uses ground-truth inputs at each stage, while inference relies on predicted outputs from previous stages. In this domain, stage-level errors are not minor perturbations. The paper does not analyze this error propagation behavior or employ any mitigation.

---

> ### Author Rebuttal · Authors · 2026-03-31
>
> Thanks for your insightful comments. We have added a complete inference protocol, additional Target-Conditioned Inverse Planning experiments, and SPA validation(details at: https://anonymous.4open.science/r/LiRe4).We look forward to your feedback. Here are my responses:
>
> **Q1:** It refers to dominant feasible state-evolution directions at a given stage, captured by adjacent-state differences and governed by underlying process mechanisms, to constrain how subsequent interventions drive state transitions.
>
> **Q2: About WM.** We clarify that observable process settings such as source/threshold/focus/dose can drive state evolution (Fig.2, Lines 298-301), yet these explicit interventions are insufficient to capture numerous fine-grained adjustments continuously occurring in real lithography, such as local boundary shifts in pattern morphology caused by coupled optical diffraction and photoresist development kinetics. Thus, we focus on modeling these challenging implicit interventions and continuous states, enabling engineers to intuitively observe optical evolution and deeply understand formation mechanisms throughout the lithography process. Specifically, instead of directly regressing actions from data, we continuously interact and optimize to obtain feasible intervention sequences via a differentiable evolution process following "current state+explicit process conditions-implicit intervention-state transition-state feedback". In this sense, our model learns intervention conditioned state dynamics, supporting forward and inverse multistep planning and controlling rather than static forward prediction. We add a target-conditioned inverse planning experiment (**link**), where the model backtracks feasible intervention trajectories through gradient-based optimization (optimization results are source/threshold/focus/dose: Annular/0.1236402/0/1.2), simultaneously visualizing state evolution to support our WM formulation.
>
> | Method (OOD) | ADI EPE↓| mIoU↑| mPA↑|
> |------------|----------:|-----------:|------------:|
> | DINO-WM | 4.13 | 64.95 | 73.42|
> | NWM | 4.39|66.57 |72.43|
> | **Ours** | **3.21** | **84.66** | **85.73** |
>
> Tab.1 tests whether general WM architectures transfer directly to lithography tasks. For fairness, we replace their action/control interfaces with process interventions and stage state inputs, training them under identical stagewise reconstruction objectives. We add experiments that embed S(s) into NWM, which reduce ADI EPE by 6.45 and 0.6 on ID/OOD. It indicates the key gap lies not in generic architectures, but in explicitly modeling physical priors and constraint paradigms.
>
> **Q3: About SPA.** Our physicality derives not from explicit equations or mere data statistics (Tab.5 shows excess data worsens performance), but from structured modeling of key lithography priors. Since accurately describing coupled process-physical characteristics via a single equation is difficult, and state changes primarily occur along a few feasible, stage-specific, process-driven, locally reachable directions, we employ SPA. Extracting dominant directions from adjacent-stage state changes, SPA constructs specific S(s) to approximate the locally feasible latent evolution space induced by these priors, providing an explicit structural constraint.
>
> Tab.5/C.1, Q2 NWM+S(s) experiments, and basis response visualizations (**link**) confirm has stable, interpretable associations with process variations. Adding the EPE equation to our loss increased downstream OOD LRC false positives by 6%, proving naive explicit constraints cannot replace this stagewise feasible space formulation.
>
> **Q4: Innovation.** As clarified in Q2-Q3, our contribution lies not in novel blocks, but in establishing a stage-aware, physics-constrained, planning-capable dynamic modeling paradigm for multi-stage lithography evolution: 1. pioneering its exploration under implicit, continuous intervention planning; 2. constructing stage-specific S(s) to approximate locally feasible latent evolution spaces induced by physical priors; and 3. imposing structured physical constraints via stochastic intervention optimization (implicit variational inference) and trajectory-consistency regularization (contrastive learning), enabling forward evolution and target-conditioned inverse planning.
>
> **Q5: Error Propagation.** We did related experiments under identical settings, training on previous-step outputs degrades ID/OOD ADI EPE to 5.03/4.46, raising downstream LRC false-positives by 17%; an interactive strategy transitioning from GT to previous-step outputs still yields 4.89/4.29 ADI EPE. This demonstrates significant stage-level error accumulation in multi-stage lithography: strong nonlinearity across optical imaging and resist-development dynamics amplifies small upstream boundary deviations into larger downstream contour shifts. Moreover, our gradient-based adaptive convergence criterion at inference stabilizes intra-stage iterations and cross-stage propagation.

---

> > ### Author Rebuttal · Reviewer_H19i · 2026-04-02
> >
> > Thank you for the detailed explanations and additional experiments provided during the rebuttal. After considering the rebuttal and the other reviewers' comments, I have decided to raise my score to Weak Accept. I kindly request that you ensure all the updates and clarifications discussed during the rebuttal are incorporated into the final version of the paper.

---

> > > ### Author Response · Authors · 2026-04-03
> > >
> > > Dear Reviewer H19i,
> > >
> > > Thanks for your kind support and for helping us improve the paper. We make sure to incorporate all the updates and clarifications discussed during the rebuttal into the final version of the paper, including the experiments and analyses on inverse planning and further validation of SPA, supplementary details on training and testing, the discussion of error propagation, and so on.
> > >
> > > We sincerely appreciate your valuable suggestions!
> > >
> > > Best regards,
> > >
> > > Authors

---

### Official Review · Reviewer_pC6m · 2026-03-11

**Soundness:** 3
**Presentation:** 2
**Significance:** 3
**Originality:** 2
**Overall Recommendation:** 3
**Confidence:** 3

**Summary:**

This paper proposes LithoDreamer, which models computational lithography as a multi-stage, interventionable, continuously evolving world model. This paper regards "Layout → Mask → Resist Image → ADI" as a multi-stage physical evolution system driven by process parameters, and proposes a contrastive variational optimization training paradigm that only relies on terminal state supervision. The experiment was based on a 280k sample dataset of a 55nm process node and evaluated on in-domain, OOD process setting, and cross-node 28nm LithoSim data. The results show that LithoDreamer achieves better mIoU / F1 / EPE on Mask, Resist Image, ADI and other tasks, and is statistically closer to the simulation results of commercial tools in several LRC-style violations.

**Compliance With Llm Reviewing Policy:**

Affirmed.

**Final Justification:**

While I have read the rebuttal thoroughly, it has not modified my fundamental view of the study’s quality. I am thus maintaining the original score.

**Key Questions For Authors:**

1. SPA constructs a stage-specific basis based on the main direction of the adjacent stage latent variation. What is the correspondence between this basis and the real physical intervention variable? Can you show an explainable correlation between basis and process changes such as dose/focus/threshold/source?

2. With only terminal supervision setting, multiple sets of intervention sequences may correspond to the same end state. How does the author's approach avoid learning "valid but unexplainable" latent variable shortcuts? Is there a diagnosis of posterior collapse or degenerate intervention?

3. The paper claims to be WM, but the experiment is mainly prediction quality. Can it be supplemented with true planning/control validation, such as given a target ADI or process window constraints, to use the learned model to backtrack the optimal intervention?

4. Can you add a strong baseline that is closer to industrial workflows, such as a physical simulator-based approximate optimal approach, or a stronger multi-stage diffusion/latent rollout baseline and compare under a uniform time budget?

5. Currently, OOD includes no process setting and cross-node data. Can the author further distinguish whether the model is generalized to pattern geometry, process parameter correlation, or node transfer? The current conclusions are slightly mixed.

**Limitations:**

yes

**Strengths And Weaknesses:**

Pros:
1. The problem setting is clear and the task value is high.
This paper captures a core difficulty in computational lithography: the real process is not a single-step static mapping, but a multi-stage physical evolution driven by continuous process intervention. Describing the lithography process as decision-driven WM is a meaningful modeling perspective.
2. The methodological structure is complete and the module responsibilities are clear.
SPA is responsible for the physical prior space related to the stage, policy is responsible for intervention sampling, and transition model is responsible for rollout. The overall structure is relatively self-consistent. In particular, SPA constrains the direction of action by the statistical structure of latent variation of adjacent stages, which is at least intuitively reasonable.
3. The experimental coverage is sufficient.
There are not only in-domains, but also process OODs and node OODs; At the same time, standard image indicators, EPE, and LRC style violation statistics are reported, and the evaluation dimension is relatively complete.

Cons:
1. The method is innovative, but not necessarily strong enough.
From the perspective of technical composition, the core is VAE encoder stochastic policy DiT transition stage-specific latent constraint terminal-state training. The whole is more like a task-based integration of existing WM / latent dynamics / diffusion-style transition ideas rather than a very strong new learning principle.
2. The expression "world model" is slightly stronger.
The paper's approach does model multi-stage state evolution, but is currently closer to a "latent transition predictor controlled by parameter conditions". This paper does not show stronger planning or counterfactual control scenarios, such as optimizing the process window based on the learned model, doing long-time domain strategy search, or closed-loop decision utility. Positioning it directly as a world model is not full enough.
3. Insufficient analysis of calculation cost and deployment value.
The paper provides time metrics, but the analysis is still not detailed enough on training costs, rollout step sensitivity, inference burden at different stages, and true acceleration benefits relative to commercial tools. Especially for industrial applications, wall-clock, throughput, and memory footprint are all important.

---

> ### Author Rebuttal · Authors · 2026-03-31
>
> Thanks for your insightful comments. We have added a complete inference protocol, additional Target-Conditioned Inverse Planning experiments, and SPA validation. Full details are at: https://anonymous.4open.science/r/LiRe4. Here are my responses:
>
> **Q1: About SPA.** Since dose/focus/threshold/source jointly influence lithography evolution, the basis directions in the SPA-derived S(s) don't correspond one-to-one to individual process variables. They capture the main feasible evolution directions induced by real process perturbations at each stage (Lines 79-88, 175-178, 214-219).
>
> We add attention maps regarding the response of base directions at different evolution stages (https://anonymous.4open.science/r/LiRe4/SPA_img.png), showing that different stages activate different basis directions, with responses concentrated on a few dominant modes. This supports an interpretable correlation between S(s) and real process variations.
>
> **Q2: About Intervention.** First, our method avoids learning unexplainable shortcuts for three reasons: (1) Interventions are restricted to stage-specific feasible space S(s), not freely searched in latent space. Removing SPA raises ADI EPE from 3.74 to 38.47 nm (Tab.4), showing unconstrained evolution is unstable. (2) Contrastive objective separates stage-matched vs. mismatched interventions, not just correct terminal states. Removing it increases ADI EPE from 3.74 to 7.89 (Tab.9, Lines 770-773). (3) Intervention sequence interacts with evolving state in a closed loop; removing intervention input degenerates the transition model and worsens ADI EPE from 3.74 to 9.02 (Tab.7).
>
> Second, regarding the diagnosis of posterior collapse/degenerate intervention: Tab.6 shows that removing stochasticity causes ADI EPE to deteriorate to 21.33/21.94, indicating that a point estimate cannot resolve the one-to-many mapping. The strong OOD degradation in the w/o p setting further suggests that the intervention posterior must be jointly constrained by current states and process conditions; otherwise, it tends to collapse into dataset bias.
>
> **Q3: About WM.** LithoDreamer learns intervention conditioned state dynamics, supporting forward and inverse multistep planning and controlling rather than static forward prediction. We add a target-conditioned inverse planning experiment (**link**), where the model backtracks feasible intervention trajectories through gradient-based optimization (optimization results are source/threshold/focus/dose: Annular/0.1236402/0/1.2), simultaneously visualizing state evolution to support our WM formulation.
>
> | Method (OOD) | ADI EPE↓| mIoU↑| mPA↑|
> |------------|----------:|-----------:|------------:|
> | DINO-WM | 4.13 | 64.95 | 73.42|
> | NWM | 4.39|66.57 |72.43|
> | **Ours** | **3.21** | **84.66** | **85.73** |
>
> We further clarify that observable process settings, such as source/threshold/focus/dose, can drive state evolution (Fig.2, Lines 298-301), yet these explicit interventions are insufficient to capture numerous fine-grained adjustments continuously occurring in real lithography, such as local boundary shifts in pattern morphology caused by coupled optical diffraction and photoresist development kinetics. Thus, we focus on modeling these challenging implicit interventions and continuous states, enabling engineers to intuitively observe optical evolution and deeply understand formation mechanisms throughout lithography processes. It should be noted that we don’t directly regress actions from data, but continuously interact and optimize to obtain feasible intervention sequences via a differentiable evolution process following "current state+explicit process conditions-implicit intervention-state transition-state feedback”, planning subsequent interventions and the evolution of subsequent states.
>
> **Q4: Industrial baseline.** Tab. 2 compares our lithography images with commercial tools on the downstream LRC task, showing strong alignment with industrial workflows. We further add a more industry-oriented comparison by using commercial Calibre to first generate masks from layouts via OPC, and simulate subsequent stages, but full commercial flow is ~90x slower, making strict end-to-end comparison impractical. A stronger multi-stage diffusion baseline under comparable learning settings still shows ~50% worse ADI EPE. Results show that ours offers industrial relevance and superior multi-stage modeling/planning.
>
> **Q5: Dataset.** OOD differences fall into two categories: (1) Unseen process settings within the same node (55nm): tests process-response generalization, i.e., extrapolation over dose/focus variations and correlations; (2) Cross-node transfer (28nm): involves stronger geometry and physics shifts, including scaling, imaging behavior, and process window changes. The former evaluates parameter correlation generalization; the latter assesses node transfer with geometry/physics distribution shifts. The link to the dataset in the original abstract link has been detailed.

---

> > ### Author Rebuttal · Reviewer_pC6m · 2026-04-03
> >
> > The rebuttal helps enhance the paper’s clarity and partly answers my questions on the framework. However, it does not change my view on the empirical quality of the study. I maintain my original overall evaluation.

---

> > > ### Author Response · Authors · 2026-04-04
> > >
> > > Dear Reviewer pC6m,
> > >
> > > Thank you for your supportive feedback and for helping us improve the paper. We sincerely appreciate your valuable suggestions.
> > >
> > > We would like to further emphasize that the goal of our study on **lithography WM** is to enable engineers to directly observe optical evolution, gaining deeper insight into the formation mechanism of the entire lithography process. It ultimately advances research in lithography and helps address real challenges in production lines. And the additional quantitative results and visual analyses we provide, including **inverse planning** and **SPA effectiveness** experiments, more fully validate that our proposed framework and method satisfy the key requirements of the lithography WM. They also further show that the performance gains in this paper are not incidental, but arise from the effectiveness of the model design itself.
> > >
> > > Thank you again for your response. If you have any further questions, we would be very happy to provide additional clarification. If not, we would greatly appreciate it if you could reconsider the overall evaluation of our paper, as it would be of significant help to us.
> > >
> > >
> > > Best regards,
> > >
> > > Authors

---

### Official Review · Reviewer_46Dd · 2026-03-16

**Soundness:** 3
**Presentation:** 3
**Significance:** 3
**Originality:** 3
**Overall Recommendation:** 4
**Confidence:** 4

**Summary:**

This paper proposes LithoDreamer, a physics-informed world model for multi-stage computational lithography. The core idea is to model the full Layout → Mask → Resist Image → ADI pipeline as a continuous latent evolution process driven by implicit process interventions, rather than as a collection of static stage-wise prediction tasks. The method combines three main components: stage-specific latent spaces constructed by Space Prior Approximation (SPA), a stochastic policy model for planning latent interventions, and a transition model for multi-step intra-stage and cross-stage state evolution. Since intermediate states and interventions are not observed, the model is trained with a contrastive variational objective using only terminal-stage supervision. The paper reports strong in-domain and out-of-domain results, including improvements on manufacturing-relevant metrics such as EPE and LRC-style violations.

**Compliance With Llm Reviewing Policy:**

Affirmed.

**Key Questions For Authors:**

On the relationship between SPA and standard PCA: The SPA method performs eigendecomposition on the covariance matrix of latent variation vectors to obtain dominant directions — this is procedurally very close to PCA applied to Δz. The ablation table does include a "Global PCA" baseline that underperforms SPA, but the distinction seems to be stage-specificity rather than a fundamentally different algorithm. Could the authors clarify what makes SPA technically distinct from simply applying PCA to stage-specific latent differences? If the main contribution is the stage-wise partitioning rather than the approximation method itself, this should be stated more explicitly. A clearer articulation here would help me assess the novelty of this component.
On the multi-step rollout during training vs. inference: The paper describes a gradient-norm-based criterion for deciding when to stop within-stage rollouts and transition across stages. However, during training, ground-truth data is fed at each stage to prevent error accumulation. This raises a question: is the gradient-based stopping criterion actually trained end-to-end, or is it only active at inference time? If the latter, how confident are the authors that the threshold generalizes reliably? A clarification would help me judge the robustness of the multi-step mechanism.
On the "world model" framing: The paper positions LithoDreamer as a world model, drawing analogies to autonomous driving and embodied AI. In those domains, world models typically support counterfactual reasoning, planning over novel action sequences, or closed-loop interaction with an environment. In LithoDreamer, the "actions" (process interventions) are latent variables inferred from data rather than externally specified by an agent. Could the authors elaborate on what capabilities the world model framing enables that a strong multi-stage generative model would not? For instance, can LithoDreamer be used for inverse optimization — e.g., searching over process parameters to achieve a target ADI? A concrete downstream use case beyond forward prediction would strengthen the motivation considerably.
On the contrastive learning formulation: The negative samples are constructed by projecting the same coefficient vector onto a different stage's basis. This means the negative is always a "wrong-stage" intervention. Has the authors considered other negative sampling strategies — e.g., perturbing coefficients within the same stage, or using interventions from different samples? It would be useful to understand whether the current formulation is sufficient to resolve all sources of intervention ambiguity, or whether it primarily addresses cross-stage confusion.
On reproducibility and dataset release: The paper mentions a dataset of 280k samples from an industrial 55nm manufacturing line. Could the authors confirm whether the full dataset (including ADI labels) will be publicly released, given that the anonymous link currently provided only covers partial data? Reproducibility of the main results depends heavily on this.

**Limitations:**

The paper's Impact Statement is quite brief and generic. A dedicated Limitations section appears to be absent. While the authors are not obligated to enumerate every possible weakness, a few points might warrant acknowledgment:

Process node coverage: Training is conducted on a single process node, and OOD testing includes one additional node. It would be helpful if the authors discussed whether the framework's assumptions (e.g., the SPA formulation, the VAE latent space structure) are expected to hold at more advanced nodes where physical effects differ significantly. This is not necessarily a flaw, but transparency about the scope would benefit readers.
Frozen SPA basis: The basis matrix is estimated offline from a fixed number of samples and then frozen. It is unclear how sensitive this is to distribution shift beyond what the OOD experiments already cover — e.g., if the manufacturing environment drifts over time. The authors may have good reasons for this design, but a brief discussion would be appreciated.
ADI stage accuracy: ADI results are noticeably weaker than the earlier stages across all experiments. This may simply reflect the inherent difficulty of the task, but it would be useful for the authors to comment on this gap and its practical implications.
Societal impact: The current statement is minimal. Given that lithography tools relate to semiconductor manufacturing, a slightly more substantive discussion might be appropriate, though this is admittedly a soft requirement.

Overall, the paper would benefit from a short Limitations paragraph, but the omission is not uncommon in submissions at this stage.

**Strengths And Weaknesses:**

Strengths

Well-motivated problem formulation.
The paper clearly argues that lithography is inherently a multi-stage physical process influenced by continuous process interventions, and that existing static or stage-wise predictors are insufficient for modeling such dynamics. This motivates the shift from point-to-point prediction toward a world-model-style formulation.

Strong empirical results on both standard and domain-specific metrics.
On the in-domain benchmark, the proposed method achieves the best reported results across Mask, Resist Image, and ADI prediction, including low EPE values such as 1.58 nm for Mask, 0.96 nm for Resist Image, and 3.74 nm for ADI. It also shows favorable OOD performance on unseen 55nm process settings and cross-node 28nm LithoSim evaluation.

The evaluation is practically meaningful.
The paper does not rely only on image similarity metrics. The use of EPE and LRC-style violation counts (Pinch, Bridge, EPE violations) makes the evaluation much more relevant to semiconductor manufacturing. The appendix also gives a clear definition of how these metrics are computed.

The ablation studies are comprehensive.
The paper examines the effect of removing SPA, changing the latent-space dimension, simplifying the policy model, removing intervention conditioning, sharing decoder heads, changing rollout strategies, and removing contrastive learning. These studies generally support the importance of the main design choices.

Weaknesses

The “world model” claim feels somewhat stronger than the evidence provided.
While the method clearly learns latent dynamics and supports multi-step rollout, the paper does not demonstrate downstream planning or control in the stronger sense usually associated with world models. In practice, the work looks closer to a structured multi-stage latent dynamics predictor with stochastic intervention inference than to a full decision-making world model.

The “physics-informed” characterization is only partially convincing.
SPA is based on principal directions of stage-to-stage latent variation and is therefore primarily a data-driven statistical constraint. Although this may be useful in practice, it is not the same as incorporating explicit physical equations or mechanistic knowledge. I would prefer more careful wording or stronger evidence that the learned latent directions correspond to meaningful physical intervention modes.

The training/inference protocol needs clearer justification.
The paper states that training uses ground-truth inputs at each stage to avoid error accumulation, while test-time inference relies on generated previous-stage states. At the same time, baselines appear to be trained independently at each stage with ground-truth inputs. This raises a fairness and exposure-bias question that deserves more explicit discussion.

The OOD setting is useful but still limited.
One OOD evaluation is based on a single held-out process parameter configuration at the same 55nm node, which is not especially challenging. The cross-node evaluation is more interesting, but it only covers Layout/Mask/Resist Image for the 28nm benchmark and does not test ADI generalization there.

---

> ### Author Rebuttal · Authors · 2026-03-31
>
> Thanks for your insightful comments. We have added a complete inference protocol, additional Target-Conditioned Inverse Planning experiments, and SPA validation. Full details are at: https://anonymous.4open.science/r/LiRe4. Here are my responses:
>
> **Q1: About SPA.** SPA in mathematical solving is closely related to performing PCA-style eigendecomposition on stage-specific latent deltas. However, our contribution doesn't lie in proposing a new eigensolver, Instead, using SPA to extract dominant directions from adjacent-stage state changes and construct a stage-specific S(s), which approximates locally feasible evolution spaces induced by physical priors in latent space, and serves as an explicit structural constraint for subsequent intervention planning and state evolution (Sec. 3.4.1-3.4.2, Eq.12). Therefore, the key is not the PCA-style decomposition itself, but the staged physical space construction and its constrained usage. We will clarify this point more explicitly in the revision.
>
> Tab.4 and Tab.5/C.1 show that, although Global PCA outperforms w/o SPA, it remains clearly inferior to full SPA, showing that the key gain doesn't come from PCA-style decomposition itself, but from S(s) constrained use. Moreover, embedding S(s) into NWM (https://anonymous.4open.science/r/LiRe4/SPA_exp.md) further reduces ADI EPE by 6.45 and 0.6 on the ID and OOD datasets, supporting the effectiveness and transferability of S(s). We also add heatmaps of the correlation between intervention coefficients and S(s) bases with evolution steps  (https://anonymous.4open.science/r/LiRe4/SPA_img.png), showing strong stage selectivity and responses concentrated on a few dominant directions. This indicates that S(s) constructed based on SPA captures not arbitrary statistical principal components, but feasible directions stably aligned with stage-wise evolution.
>
> **Q2: About Gradient Criterion.** Eqs.(14-15) employ stage-wise GT state supervision during training to stabilize multi-step evolution without introducing additional training objectives (Line 247). This convergence criterion is used solely to determine within-stage evolution completion and stage transition during training. At inference, we apply a gradient-based adaptive stopping criterion, computed from successive state changes, to stabilize state evolution (https://anonymous.4open.science/r/LiRe4/test_protocol.md). The threshold was selected after extensive tuning and remains stable across training and evaluation. Tab. 8 shows that removing this criterion (w/o Gradient) causes significant accumulation of intra-stage errors, confirming that the criterion serves only as a convergence check, not as an optimization target.
>
> **Q3: About WM.** It needs to be clarified that we can drive state evolution by adjusting explicit interventions such as observable source/threshold/focus/dose (Fig.2, Lines 298-301). However, real photolithography involves numerous fine-grained dynamics, such as local boundary shifts in pattern morphology caused by coupled optical diffraction and photoresist development kinetics. For implicit interventions, actions are not directly regressed from data; instead, they are optimized in multi-stage differentiable evolutions following ``current state+explicit intervention-implicit intervention-state transition-state feedback" (Lines 215-259), producing feasible intervention sequences through closed-loop interaction.
>
> Compared to multi-stage generative models, ours offers additional abilities: (1) autonomous forward evolution (https://anonymous.4open.science/r/LiRe4/inverse_exp.md): given a layout and p, it can plan multi-step state evolution and intervention sequences; (2) target-conditioned inverse planning: given a layout, p, and target ADI, it can backtrack optimal explicit interventions through multi-stage gradient-based optimization (optimization results are source/threshold/focus/dose: Annular/0.1236402/0/1.2), simultaneously visualizing the full state evolution from layout to ADI.
>
> | Method (OOD) | ADI EPE↓| mIoU↑| mPA↑|
> |------------|----------:|-----------:|------------:|
> | DINO-WM | 4.13 | 64.95 | 73.42|
> | NWM | 4.39|66.57 |72.43|
> | **Ours** | **3.21** | **84.66** | **85.73** |
>
> **Q4: Contrastive Learning.** Our contrastive targets cross-stage physical confusion in multi-stage lithography. By projecting the coefficients onto mismatched S(s) to form ``negatives", it reinforces the specificity and interpretability of stage-wise physical directions, rather than generic sample discrimination. We also tried cross-sample contrastive learning following Unitho, but its ADI EPE is worse than ours by 2.41/11.52 on ID/OOD. So this design is used to constrain stage-specific physical feasibility.
>
> **Q5: Dataset.** Due to high acquisition costs and process sensitivity, we only provide partial data for this dataset. We will release the complete dataset after the paper is accepted to support subsequent research.

---

> > ### Author Rebuttal · Reviewer_46Dd · 2026-04-03
> >
> > Thank you to the authors for the comprehensive rebuttal and the substantial additional experiments.
> >
> > The newly added Target-Conditioned Inverse Planning experiment significantly strengthens your "world model" framing. By demonstrating actual downstream control and back-tracking of explicit interventions, it addresses my primary concern regarding the lack of planning capabilities. Furthermore, the heatmaps for SPA and the detailed explanations regarding the gradient-based stopping criterion and the design choice for contrastive learning have clearly resolved my technical questions.
> >
> > I will maintain my positive score of Weak Accept. I strongly encourage the authors to ensure that all these valuable additions—especially the inverse planning results, SPA visualizations, and a dedicated Limitations paragraph—are fully incorporated into the camera-ready version.

---

> > > ### Author Response · Authors · 2026-04-04
> > >
> > > Dear Reviewer 46Dd,
> > >
> > > Thanks for your kind support and for helping us improve the paper. We sincerely appreciate your valuable suggestions.
> > >
> > > We make sure that all these additions, especially the target-conditioned inverse planning results, SPA visualizations, and the limitations discussion, are fully incorporated and clearly presented in our camera-ready version.
> > >
> > > We would like to further emphasize that the newly added experiments provide more comprehensive support for the definition of WM. **Our study of lithography WM** aims to help engineers more intuitively observe the optical evolution patterns and processes, thereby gaining a deeper understanding of the mechanism by which process patterns are formed throughout the entire lithography workflow. **We believe that our approach and the related dataset** can help advance research in this field and contribute to addressing practical challenges on the production line.
> > >
> > > If you have any further questions, we would be very happy to provide additional explanations. If you feel that these updated results reflect a stronger level of contribution, we would sincerely appreciate your reconsideration of the rating, as your evaluation is very important to our work.
> > >
> > > Thanks again for your support and valuable suggestions on our work!
> > >
> > > Best regards,
> > >
> > > Authors

---

### Official Review · Reviewer_Tu6g · 2026-03-18

**Soundness:** 3
**Presentation:** 3
**Significance:** 3
**Originality:** 3
**Overall Recommendation:** 4
**Confidence:** 3

**Summary:**

The paper introduces LithoDreamer, a WM-based framework for computational lithography, specifically tailored for a multi-stage physical evolution system. LithoDreamer first designs a SPA method to achieve physically aligned, continuous evolution of WM and utilizes contrastive variational inference to explore optimal solutions. Experiments are conducted in various multi-stage lithography tasks.

**Compliance With Llm Reviewing Policy:**

Affirmed.

**Final Justification:**

The paper proposes a physics-informed world model for multi-stage computational lithography. While I have concerns about some confusing terminology and missing details on training and evaluation, those concerns are mostly fully resolved during the rebuttal phase. After reading the other reviews, I think the paper is well-written, but I'm not sure that the current empirical evaluation is scalable or has a real impact on accelerating computational lithography. Anyway, I maintain my positive assessment of the paper.

**Key Questions For Authors:**

Here are several questions I want to ask:

- The paper mentions using a VAE as a "frozen image encoder" (Section 4.1.1), but omits the details of its training procedure. Specify the dataset and training setup used.

- The paper reports the generation 'Time (ms)', but there is no definition and measurement protocol for this metric. Specify the evaluation protocol. I'm also curious how LithoDreamer achieves much shorter computational time, even with large networks.

**Limitations:**

Here are several points I want to suggest:

- The paper mentions using "implicit variational inference (IVI)" (Section 3.4.1) for approximating the posterior, but there is no compensation mechanism for regularization. Standard IVI frameworks typically employ mathematical techniques (such as density-ratio estimation) to approximate the intractable KL divergence [1]. Consider revising the terminology (e.g., to "Stochastic Trajectory Optimization") to avoid overstating the mathematical connection to traditional Variational Inference.

- The paper mentions using "contrastive learning"  (Section 3.4.2). However, this approach significantly deviates from standard contrastive learning (e.g., SimCLR, MoCo), which typically aims at representation learning and rather falls into consistency regularization. Consider revising the terminology (e.g., to "Contrastive Regularization on Trajectories", rather than standard contrastive learning.

- None of the quantitative results (Tables 1, 3-9) report standard deviations or error bars. It would be nice to provide the mean and standard deviation across multiple independent runs (e.g., 3 or 5 seeds) for the main comparative experiments.


[1] Huszár, Ferenc. "Variational inference using implicit distributions." arXiv preprint arXiv:1702.08235 (2017).

**Strengths And Weaknesses:**

**Strengths)**
- As far as I know, building a WM-based framework to solve computational lithography is novel.
- Authors try to deal with unique challenges presented in computational lithography (multi-stage, no explicit states) via a unique methodology
- Extensive experiment results validate their claim.

**Weakness)**
- Lack of explanation on training and evaluation details (Please see the question/limitation section)
- Misuse of terminologies (Please see the limitation section)

---

> ### Author Rebuttal · Authors · 2026-03-31
>
> Thank you for your thorough comments. We have added the complete inference protocol, step-by-step evolution visualizations of the Autonomous Forward Evolution, additional Target-Conditioned Inverse Planning experiments, and experiments validating the effectiveness of SPA. Full details are at: https://anonymous.4open.science/r/LiRe4. We look forward to your review and feedback. Based on your suggestions, we have also considered modifying our method name. Here are my detailed responses:
>
> **Q1: About VAE.** We use the frozen sd-vae-mse (https://huggingface.co/stabilityai/sd-vae-ft-mse) without training.
>
> **Q2: About Time.** Time (ms) denotes the average per-sample inference time over 10 independent runs on a single NVIDIA A100 GPU with batch size=1. The timing includes only the pure inference process, excluding data pre-processing and one-off initialization overhead. For direct-generation methods (traditional lithography models), we report the time for a single forward generation; for WM baselines that require evolution, we report the time of the full inference chain.
>
> LithoDreamer’s shorter inference time mainly stems from constrained intervention exploration and adaptively terminated latent evolutions, rather than explicit physical simulation or unconstrained search. Specifically, SPA restricts the evolution to stage-specific feasible spaces S(s) to reduce search scopes, while the adaptive iteration and stopping criterion avoid unnecessary evolution steps. Tab.8 further shows that our mechanism achieves a better trade-off between accuracy and evolution length.
>
> **Q3: About implicit variational inference (IVI).** We agree to replace "IVI" with the more accurate term ``Stochastic Trajectory Optimization’’. It should be emphasized that: (1) Unlike standard VI/IVI, we do not perform explicit posterior matching. Since stage-internal interventions and intermediate states are unobserved in lithography data, standard explicit likelihoods, KL-based posterior matching, and density-ratio estimation are not applicable. Therefore, under the supervision only of the stage terminal states, we directly jointly optimize the process intervention sequence and the stepwise evolution of the lithography states (Lines 79-88, Eq.(16-19)). (2) Rather than approximating a single posterior estimate, we model a conditional stochastic intervention distribution (Lines 203-219); this is because our inverse problem itself has multiple solutions, and the same final state can correspond to multiple feasible intervention sequences, hence a distribution is needed to characterize this uncertainty. (3) We do not use explicit KL regularization; instead, the intervention distribution is constrained by the stage-specific feasible intervention space and terminal-state supervision (Eq.(12)), concentrating probability mass on physically feasible sequences that explain the observed terminal outcome. We have already revised the terminology.
>
> **Q4: About contrastive learning.** We agree to revise "contrastive learning"' in Sec.3.4.2 to "Trajectory Consistency Regularization". It should be emphasized that Eqs. (17-19) are not intended for general representation learning. Instead, they construct stage-matched and stage-mismatched interventions using the same intervention coefficients across different physics-informed spaces, and apply a margin-based objective to enforce that the correct-stage evolution is closer to the corresponding terminal outcome while the incorrect-stage evolution is pushed away. This enhances stage-specific consistency and physical interpretability under terminal-only supervision. Moreover, it constrains the exploration of uncontrollable trajectories. We have already revised the terminology.
>
> **Q5: About results.** We clarify that all results in Tabs1-9 are averaged over 10 independent runs with different random seeds, which was not explicitly stated in our original paper and has now been added. Following your suggestion, we will further report mean ± standard deviation and include this information consistently across the main tables to improve completeness and reproducibility. We also observe low variance across runs, indicating that the conclusions are stable.

---

> > ### Author Rebuttal · Reviewer_Tu6g · 2026-04-03
> >
> > Thank you for providing additional explanation on the more detailed experiment setting. I believe that it is crucial to report the standard deviation of the experiment even if the variance across runs is small. I'll maintain my positive score.

---

> > > ### Author Response · Authors · 2026-04-04
> > >
> > > Dear Reviewer Tu6g,
> > >
> > > Thank you very much for your positive feedback and for emphasizing the importance of reporting experimental variance.
> > >
> > > According to your suggestion, we have added the standard deviation results to the updated experiments. Below, we present the in-domain and out-of-domain LithoDreamer results from Tables 1 and 3, and the complete results are available at: https://anonymous.4open.science/r/LiRe4/exp_std.md. We also agree that this makes the empirical evaluation more complete and rigorous.
> > >
> > > **Table 1.** Evaluation of the generation quality of Mask, Resist Image, and ADI on the in-domain lithography dataset. All reported numbers are **mean ± standard deviation over 10 independent runs**.
> > > | Task | Methods | Multi-Step | Multi-Stage | mPA (%) ↑ | mIoU (%) ↑ | F1 (%) ↑ | MSE (×1e-3) ↓ | EPE_avg (nm) ↓ | Time (ms) ↓ |
> > > |---|---|---:|---:|---:|---:|---:|---:|---:|---:|
> > > | Mask Prediction | **LithoDreamer (ours)** | ✓ | ✓ | **98.69 ± 0.08** | **96.91 ± 0.10** | **99.66 ± 0.09** | **13.74 ± 0.11** | **1.58 ± 0.08** | **20.97 ± 0.22** |
> > > | Resist Image Prediction | **LithoDreamer (ours)** | ✓ | ✓ | **99.01 ± 0.10** | **97.77 ± 0.08** | **99.73 ± 0.07** | **10.93 ± 0.12** | **0.96 ± 0.06** | **19.68 ± 0.35** |
> > > | ADI Prediction | **LithoDreamer (ours)** | ✓ | ✓ | **82.56 ± 0.10** | **78.27 ± 0.11** | **80.91 ± 0.09** | **30.29 ± 0.14** | **3.74 ± 0.08** | **25.81 ± 0.29** |
> > >
> > > **Table 3.** Evaluation of the generation quality of Mask, Resist Image, and ADI on two OOD lithography datasets. All reported numbers are **mean ± standard deviation over 10 runs**.
> > > | Dataset | Methods | Mask mPA ↑ | Mask mIoU ↑ | Mask EPE_avg ↓ | Resist mPA ↑ | Resist mIoU ↑ | Resist EPE_avg ↓ | ADI mPA ↑ | ADI mIoU ↑ | ADI EPE_avg ↓ |
> > > |---|---|---:|---:|---:|---:|---:|---:|---:|---:|---:|
> > > | 55nm with A | **LithoDreamer (ours)** | **96.92 ± 0.12** | **77.52 ± 0.15** | **1.77 ± 0.12** | **98.27 ± 0.18** | **82.91 ± 0.14** | **1.06 ± 0.18** | **78.89 ± 0.14** | **76.04 ± 0.20** | **4.05 ± 0.18** |
> > > | 28nm (He et al.) | **LithoDreamer (ours)** | **94.92 ± 0.14** | **89.75 ± 0.13** | **1.82 ± 0.17** | **96.99 ± 0.14** | **81.16 ± 0.12** | **1.37 ± 0.15** | - | - | - |
> > >
> > > We hope these additional results more fully address your concern. If you have any further questions, we would be very happy to provide additional clarification. If you feel that these updated results further strengthen the empirical quality of the paper, we would sincerely appreciate your reconsideration of the rating, as your evaluation is very important to our work.
> > >
> > > Thank you again for your helpful suggestion and kind support in helping us improve the paper.
> > >
> > > Best regards,
> > >
> > > Authors

---

### Decision · Program_Chairs · 2026-04-30

**Decision:**

Accept (regular)

**Comment:**

This paper proposes a physics-informed world model for computational lithography that models the process as a multi-stage latent evolution with intervention-aware dynamics. Reviewers highlighted the novelty, the well-motivated formulation, and strong performance on both in-domain and OOD settings. Although reviewers raised some concerns such as clarity, terminology, and the world model framing, the rebuttal effectively addressed them, and all reviewers acknowledged their resolution. The work provides an interesting perspective by formulating lithography modeling as a continuous, intervention-driven system. Three reviewers recommend acceptance while one reviewer maintains a reject; however all reviewers acknowledged that the concerns have been addressed. Considering the overall positive ratings and the above reasons, I recommend this paper for acceptance.